# Replication intermediates that escape Dna2 activity are processed by Holliday junction resolvase Yen1

Gizem Ölmezer[1,2], Maryna Levikova[3,*], Dominique Klein[1,*], Benoît Falquet[1,2], Gabriele Alessandro Fontana[1], Petr Cejka[3] & Ulrich Rass[1]

Cells have evolved mechanisms to protect, restart and repair perturbed replication forks, allowing full genome duplication, even under replication stress. Interrogating the interplay between nuclease-helicase Dna2 and Holliday junction (HJ) resolvase Yen1, we find the Dna2 helicase activity acts parallel to homologous recombination (HR) in promoting DNA replication and chromosome detachment at mitosis after replication fork stalling. Yen1, but not the HJ resolvases Slx1-Slx4 and Mus81-Mms4, safeguards chromosome segregation by removing replication intermediates that escape Dna2. Post-replicative DNA damage checkpoint activation in Dna2 helicase-defective cells causes terminal G2/M arrest by precluding Yen1-dependent repair, whose activation requires progression into anaphase. These findings explain the exquisite replication stress sensitivity of Dna2 helicase-defective cells, and identify a non-canonical role for Yen1 in the processing of replication intermediates that is distinct from HJ resolution. The involvement of Dna2 helicase activity in completing replication may have implications for *DNA2*-associated pathologies, including cancer and Seckel syndrome.

[1] Friedrich Miescher Institute for Biomedical Research, Maulbeerstrasse 66, Basel CH-4058, Switzerland. [2] University of Basel, Petersplatz 10, Basel CH-4003, Switzerland. [3] Institute of Molecular Cancer Research, University of Zurich, Winterthurerstrasse 190, Zurich CH-8057, Switzerland. * These authors contributed equally to this work. Correspondence and requests for materials should be addressed to U.R. (email: ulrich.rass@fmi.ch).

Duplication of the genome requires the passage of DNA replication forks along the entire length of every chromosome. If segments of DNA remain unreplicated, physical links between the nascent sister chromatids persist, which can lead to aberrant chromosome segregation[1]. Replication fork collapse, characterized by replisome inactivation and DNA breakage, induces recombinogenic DNA lesions and gross chromosomal instability[2]. Consistently, replication stress, which increases the risk of replication fork stalling, arrest, and collapse, has been recognized as a driver in cancerogenesis[3]. Cells respond to replication stress by activating the S phase checkpoint, which triggers a cascade of downstream events aimed at preserving the replication machinery at troubled replication forks until DNA synthesis can resume[4]. Replication restart also involves fork remodelling, nucleolytic processing of stalled replication intermediates and homologous recombination (HR) reactions[5–7]. Thus, full genome duplication and proper chromosome segregation is dependent on a complicated network of replication and repair proteins that remains incompletely understood.

A protein implicated in multiple aspects of DNA replication and repair is the conserved nuclease-helicase Dna2. Essential in yeast[8], *DNA2* is required for embryonic development in mice[9], and its downregulation leads to chromosomal instability[10–14]. The enzymatic activities of Dna2 reside in a RecB-like nuclease domain[15] with single-stranded DNA (ssDNA)-specific endonuclease activity[16], and a C-terminal superfamily 1 (SF1) helicase domain[8]; in yeast, Dna2 has an additional, unstructured N-terminal domain that serves a redundant function in S phase checkpoint activation[17].

The nuclease activity of Dna2, in particular, has been linked with a number of molecular pathways. *In vitro*, Dna2 cuts DNA 5′-flaps bound by replication protein A (RPA), and it has been proposed that this activity might facilitate Okazaki fragment maturation by mediating the removal of occasional long 5′-flaps, which might attract RPA and become refractory to cleavage by Rad27 (FEN1 in human)[18,19]. During DNA double-strand break repair, the Dna2 nuclease degrades the 5′-terminated single strand unwound by the Sgs1 helicase (BLM, Bloom syndrome protein in human), promoting DNA end-resection and HR redundantly with Exo1 (ref. 20). Similarly, the Dna2 nuclease has been implicated in *Schizosaccharomyces pombe* in the processing of stalled replication fork intermediates through degradation of the regressed DNA branch emanating from reversed replication forks as the newly synthesized DNA strands become displaced and anneal with one another to form a chicken-foot structure[21,22]. An analogous reaction, mediated by the DNA2 nuclease in conjunction with Werner's syndrome helicase WRN, promotes replication restart in human cells[23], while failure to properly control DNA2-mediated DNA resection at stalled forks leads to excessive DNA degradation and genome instability[24,25].

The physiological role of the Dna2 helicase activity, as opposed to the nuclease activity, has remained unclear. There is currently no evidence that the helicase activity contributes to the degradation/resection of DNA ends at reversed forks or DNA double-strand breaks. Interestingly, a number of Dna2 mutants affected within the conserved SF1 helicase motifs I-VI confer growth defects accompanied by sensitivity to the DNA alkylating agent methyl methanesulfonate (MMS)[14,26]. This phenotype is not generally shared with mutants affected in the N-terminal domain[26] or nuclease domain[27], indicating that Dna2 helicase-specific functions in the repair of DNA damage or in the response to damage-induced replication stress exist.

Intriguingly, a genetic screen[28] uncovered a synthetic sick interaction, characterized by slow growth, between *dna2-2*, an allele that encodes a Dna2 variant with a single amino acid change (R1253Q) in the helicase domain[14], and structure-specific *RAD2/XPG* superfamily nuclease *YEN1*, indicating a potential functional interplay. Yen1, and its human orthologue GEN1, are Holliday junction (HJ) resolvases[29]. These enzymes are best known for their role in processing late HR intermediates, such as fully double-stranded DNA (dsDNA) four-way HJ junctions[30]. Eukaryotes use three conserved HJ resolvases, Yen1/GEN1, Mus81-Mms4/human MUS81-EME1 and Slx1-Slx4/human SLX1-FANCP to remove recombination intermediates that form during replication-associated DNA repair processes in mitotic cells[31]. Mounting evidence suggests that Mus81-Mms4/MUS81-EME1 also targets unproductive replication intermediates, effectively breaking stalled replication forks to allow HR-dependent replication restart or repair[32]. In human cells, MUS81-EME1 promotes the expression of chromosomal fragile sites, which is thought to represent controlled breakage of underreplicated DNA at the time of mitosis to limit sister chromatid non-disjunction[33,34]. At present, there is no evidence for a similar role of Yen1/GEN1 in targeting replication—rather than recombination—intermediates.

Here, we analyse the interplay between Dna2 and Yen1 to reveal new aspects of the cellular response to replication stress. We find that the Dna2 helicase activity acts on replication fork stalling, promoting full genome duplication along a pathway parallel to HR-mediated replication fork recovery. If the Dna2 helicase fails to respond properly to stalled replication forks, replication intermediates remain and give rise to post-replicative chromosomal DNA links that preclude chromosome segregation. Resolution is uniquely dependent on the actions of Yen1, which identifies a first non-redundant function of Yen1 in protecting cells from mitotic catastrophe after replication stress.

## Results

**Dna2[R1253Q] is helicase defective and nuclease proficient**. Using budding yeast, Campbell and colleagues have conducted a large-scale genetic screen[28] using *dna2-2* (R1253Q) and nuclease-defective allele *dna2-1* (P504S)[14], which identified 37 synthetic sick/synthetic lethal interactions, predominantly with genes involved in DNA replication and repair. Many interactions were shared between *dna2-2* and *dna2-1*, but a synthetic sick interaction with *YEN1* was unique to *dna2-2*. Dna2[R1253Q] is affected at an invariant arginine in helicase motif IV of the SF1 helicase domain, suggesting that the Dna2 helicase and Yen1 may function in related pathways. However, since single amino acid changes in Dna2 have been described that impact both the nuclease and ATPase/helicase activities[35], and because Dna2[R1253Q] has never been isolated and analysed biochemically, we first assessed directly the mutant protein's ATPase/helicase and nuclease activities. Dna2[R1253Q] was purified to near-homogeneity following overexpression in *Saccharomyces cerevisiae* (Fig. 1a,b), and tested alongside wild-type Dna2, and well-established[35] nuclease-dead and helicase-dead variants, Dna2[E675A] and Dna2[K1080E], respectively.

When wild-type Dna2 was incubated with 5′-tailed DNA, the activity of the ATPase/helicase domain was readily detected, before the potent Dna2 nuclease degraded the ssDNA tails, so that the ATPase was no longer stimulated and ATP hydrolysis subsided; the nuclease-dead variant Dna2[E675A] exhibited persistent ATPase activity[35,36] (Fig. 1c). In contrast to wild-type and Dna2[E675A], Dna2[R1253Q] showed no ATPase activity, and was indistinguishable from previously characterized[35,36] ATPase/helicase-dead variant Dna2[K1080E] (Fig. 1c and Supplementary Fig. 1). *In vitro*, Dna2 exhibits ssDNA-specific nuclease activity on 5′-tailed or 3′-tailed DNA substrates, while RPA stimulates its nuclease and enforces 5′-3′ directionality,

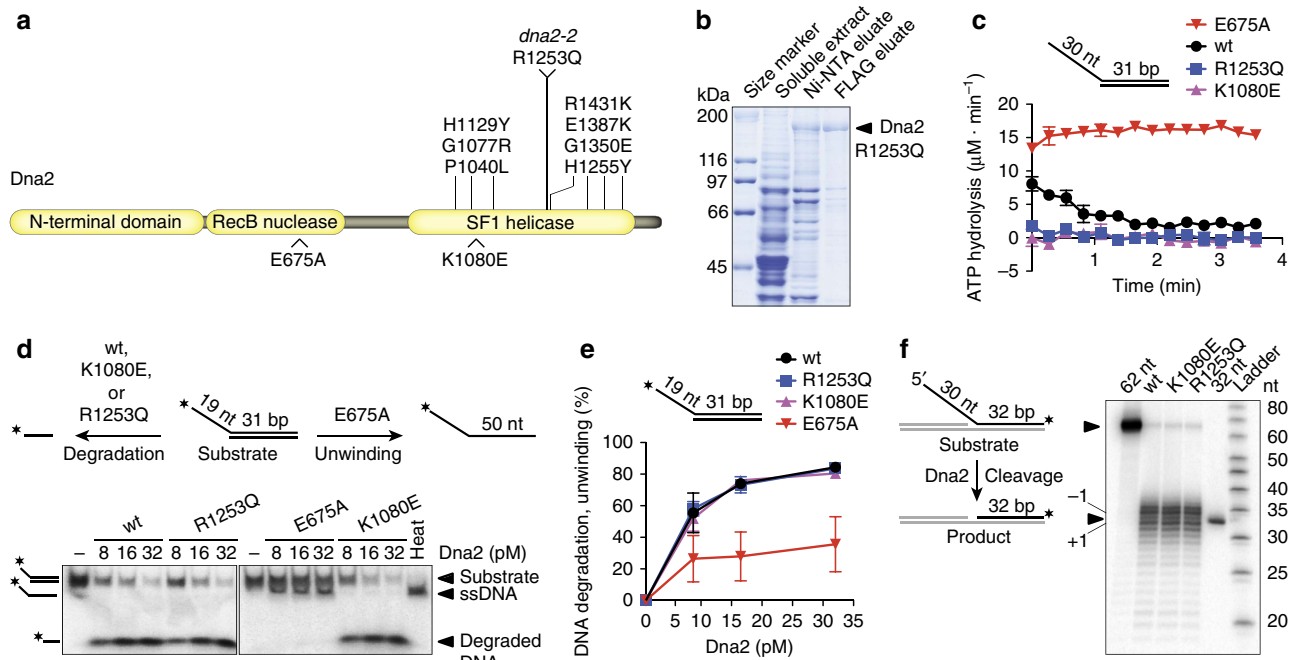

**Figure 1 | Biochemical analysis of Dna2 variant R1253Q. (a)** Domain structure of *Saccharomyces cerevisiae* nuclease-helicase Dna2. *Above*, single amino acid changes within the helicase domain of Dna2 that result in MMS sensitivity, including R1253Q, encoded by *dna2-2*. *Below*, position of mutations E675A and K1080E, which have been shown to inactivate the Dna2 nuclease or helicase activity, respectively. **(b)** Dna2$^{R1253Q}$ carrying 6 × His and FLAG tags was expressed in *S. cerevisiae* and purified by affinity chromatography using nickel-nitrilotriacetic acid (Ni-NTA)-agarose and anti-FLAG affinity gel. Fractions were analysed by polyacrylamide gel electrophoresis followed by Coomassie blue staining. **(c)** Kinetics of ATP hydrolysis by wild-type (wt) Dna2 and indicated variants (all 4 nM) in the presence of a 5′-tailed DNA substrate (1 μM nucleotides). Data are presented as mean values ± s.e.m. ($n = 2$). **(d)** Processing of 5′-tailed DNA by Dna2. The panel shows a representative 10% polyacrylamide gel with reaction products after incubation of the DNA substrate with the indicated DNA2 variants and RPA (16.8 nM). *, position of the $^{32}$P-label on the DNA. Heat, heat-denatured DNA substrate. **(e)** Quantification of experiments such as those shown in **d**. Data are presented as mean values ± s.e.m. ($n = 2$). **(f)** 5′-flap cleavage by Dna2 variants (all 2 nM) in the presence of RPA (30 nM). Reaction products were separated on a 20% polyacrylamide denaturing urea gel. Cleavage at the base of the flap produces a radiolabelled fragment of 32 nt.

which is likely the relevant polarity *in vivo*[20]. In the presence of RPA, Dna2$^{R1253Q}$ degraded 5′-tailed DNA in a manner similar to wild-type Dna2, showing that the R1253Q mutation does not interfere with the nuclease activity (Fig. 1d). In line with the observed lack of ATPase activity, we did not find evidence of DNA unwinding by Dna2$^{R1253Q}$, an activity that was readily detected for nuclease-deficient mutant Dna2$^{E675A}$ (Fig. 1d). Quantification of the nuclease/helicase assays showed that Dna2$^{R1253Q}$ was as efficient as wild-type and Dna2 helicase mutant K1080E in degrading 5′-tailed DNA (Fig. 1e). Finally, and in accord with previous studies using other Dna2 helicase mutants[37,38], Dna2$^{R1253Q}$ was fully proficient in removing 5′-flaps from dsDNA by cleavage at the flap base, in a reaction that mimics the potential role of Dna2 in Okazaki fragment processing (Fig. 1f). These results show that the *dna2-2* allele confers a helicase-specific defect and does not impinge on the activity of the Dna2 nuclease.

**Checkpoint activation and loss of YEN1 impair *dna2-2* cells.** Having established that the R1253Q mutation selectively inactivates the helicase activity of Dna2, we introduced the *dna2-2* allele into cells to investigate the effect of Dna2 helicase deficiency *in vivo*. While Dna2 protein levels were unaffected (Supplementary Fig. 2a), the R1253Q mutation caused MMS sensitivity, as expected for *dna2-2* cells[14] (Supplementary Fig. 2b). Under unperturbed conditions, the *dna2-2* strain exhibited a plating efficiency similar to wild-type. In contrast, viability dropped sharply for the *dna2-2 yen1Δ* double mutant to ∼35%

of wild-type levels (Fig. 2a). Doubling time measurements revealed that the *dna2-2* mutation was associated with a mild slow growth phenotype, extending doubling times by ∼10 min (103 min versus 92 min for wild-type). On deletion of *YEN1*, the growth phenotype was much more severe, with an increase in doubling time of ∼50 min for the double mutant (143.5 min). Consistent with previous results[28], we did not observe a synthetic growth defect when *YEN1* was deleted in Dna2 nuclease-mutant *dna2-1* cells (data not shown), indicating that the genetic interaction between *YEN1* and *DNA2* relates specifically to the Dna2 helicase activity. Contrary to a reported temperature-dependent lethal interaction between *YEN1* and *DNA2* (ref. 28), we found double mutant cells were viable at elevated temperature (37 °C) (Supplementary Fig. 2c), although doubling times for *dna2-2* and *dna2-2 yen1Δ* were further increased by ∼20 and ∼5 min, respectively.

Microscopic inspection of exponentially growing *dna2-2* cultures revealed an accumulation of cells in G2/M phase of the cell cycle, and this effect was further accentuated on deletion of *YEN1*. Morphological examination showed that *dna2-2* cultures contained ∼4% large dumbbell-shaped cells. In *dna2-2 yen1Δ* cultures, this sub-fraction was more extensive, accounting for ∼8% of cells, and ∼5% of cells exhibited morphological changes such as bud elongation and the formation of short chains of elongated cells (Fig. 2b). Finally, the vast majority of G2/M cells within the *dna2-2* and *dna2-2 yen1Δ* cultures (≥70%), but not within wild-type or *yen1Δ* cultures, contained unsegregated nuclear DNA positioned near the bud neck (Fig. 2b).

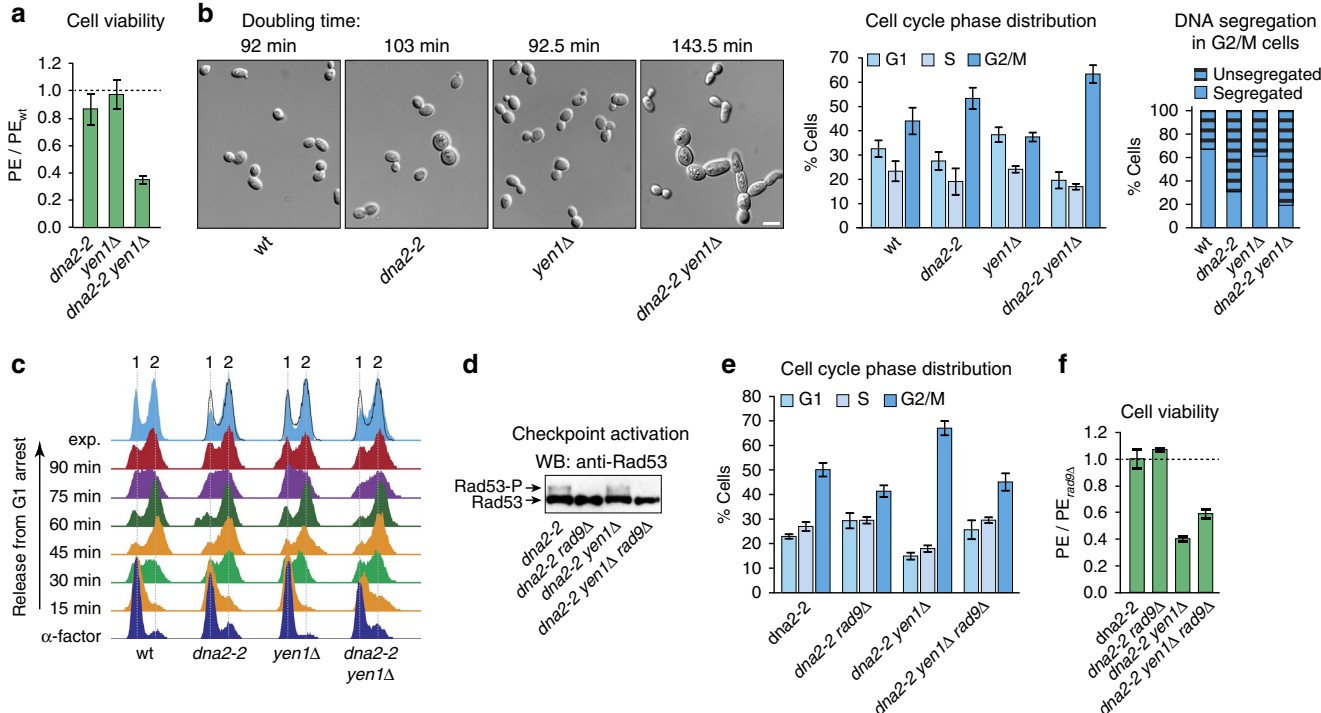

**Figure 2 | Dna2 helicase-defective cells suffer dual growth inhibition by checkpoint activation and loss of YEN1.** (**a**) Cell viability of the indicated strains assessed by colony outgrowth. The mean plating efficiency (PE) ± s.e.m. ($n = 3$) is presented relative to wild-type. (**b**) Microscopic analysis of the indicated strains growing exponentially in rich medium. *Left*, representative images (DIC) showing morphological changes associated with Dna2 helicase dysfunction and loss of Yen1. Average doubling times are given ($n = 3$). Scale bar, 5 μm. *Centre*, mean distribution of G1, S, and G2/M cells ± s.e.m. ($n = 4$). *Right*, DNA segregation in G2/M cells as determined by DAPI-staining. An average of 200 cells were scored per strain. (**c**) Flow cytometric analysis of the indicated strains synchronized in G1 using α-factor and released into YPAD. The position of cells with 1 and 2 N DNA content is indicated. Topmost tracks are asynchronous cultures overlaid with the outline of the wild-type profile, showing that Dna2 helicase-defective strains accumulate cells with a 2 N content over time. (**d**) Western blot analysis showing chronic low-level DNA damage checkpoint activation in Dna2 helicase-defective cells as indicated by Rad53 phosphorylation (Rad53-P). (**e**) Effect of DNA damage checkpoint disruption by deletion of *RAD9* on the distribution of G1, S, and G2/M cells in exponentially growing cultures of the indicated strains. Data presented as mean distribution ± s.e.m. ($n = 3$). (**f**) Cell viability of the indicated strains assessed by colony outgrowth. The mean plating efficiency ± s.e.m. ($n = 3$) is presented relative to a *rad9Δ* control.

While an accumulation of G2/M cells occurred during exponential growth, analysis by flow cytometry showed that *dna2-2* and *dna2-2 yen1Δ* cells progressed through a single cell cycle with apparently normal kinetics on synchronous release into S phase after α-factor pheromone-induced G1 arrest (Fig. 2c). This indicates that the helicase activity of Dna2 is largely dispensable for bulk DNA synthesis, but that Dna2 helicase-deficient cells have a tendency to arrest at the G2/M transition, as noted previously[14]. Importantly, replication and the G2/M transition phenotype were unaffected by the presence or absence of Yen1. The accumulation of G2/M cells may result from elevated levels of stochastic DNA damage, since we detected low-level phosphorylation of the checkpoint kinase Rad53 in *dna2-2* and *dna2-2 yen1Δ* cells in unperturbed conditions, which was suppressed on deletion of the DNA damage checkpoint mediator *RAD9* (Fig. 2d). Moreover, the levels of G2/M cells in either strain were much reduced in the absence of *RAD9*, and bud-elongation and cell-chain formation was no longer observed (Fig. 2e and data not shown). The extended doubling times for the *dna2-2* and *dna2-2 yen1Δ* strains were reduced on *RAD9* deletion, albeit not to wild-type levels (96 and 122 min, respectively). Significantly, the viability of both *dna2-2* and *dna2-2 rad9Δ* cells was indistinguishable from the *rad9Δ* control, whereas the severe reduction of viability we had observed on loss of *YEN1* in the *dna2-2* background was only mildly suppressed by *RAD9* deletion (Fig. 2f). Together these data suggest that growth defects in *dna2-2 yen1Δ* cells arise from two separate sources: (1)

Dna2 helicase dysfunction causes cells to accumulate DNA lesions during unperturbed growth, triggering Rad9-dependent DNA damage checkpoint activation and a delay at the G2/M transition. (2) Yen1 cannot prevent these defects, so that they manifest themselves similarly in *dna2-2 yen1Δ* double mutant and *dna2-2* single mutant cells. Yet, the absence of Yen1 is toxic to *dna2-2* cells, indicating that Yen1 acts downstream, resolving a catastrophic DNA metabolic event that ensues when the Dna2 helicase is non-functional.

**Yen1 is critical in *dna2*-2 cells under replication stress.** Dna2 helicase deficiency sensitizes cells to DNA alkylating agent MMS[14,26] (Supplementary Fig. 2b). To test whether loss of Yen1 has an additional effect on the MMS sensitivity of Dna2 helicase-defective cells, we exposed *dna2-2 yen1Δ* cells to increasing amounts of the drug. As expected[39], loss of *YEN1* alone did not result in overt MMS sensitivity. In contrast, *dna2-2 yen1Δ* cells proved to be several orders of magnitude more sensitive than *dna2-2* cells in drop assays (Fig. 3a). This phenotype was not restricted to MMS, and similar results were obtained with topoisomerase I poison camptothecin (CPT) and ribonucleotide reductase inhibitor hydroxyurea (HU). These drugs have disparate mechanisms of action, but their effects (DNA damage, accumulation of trapped Top1 cleavage complexes throughout the genome and nucleotide depletion, respectively) all inhibit the progression of replication forks,

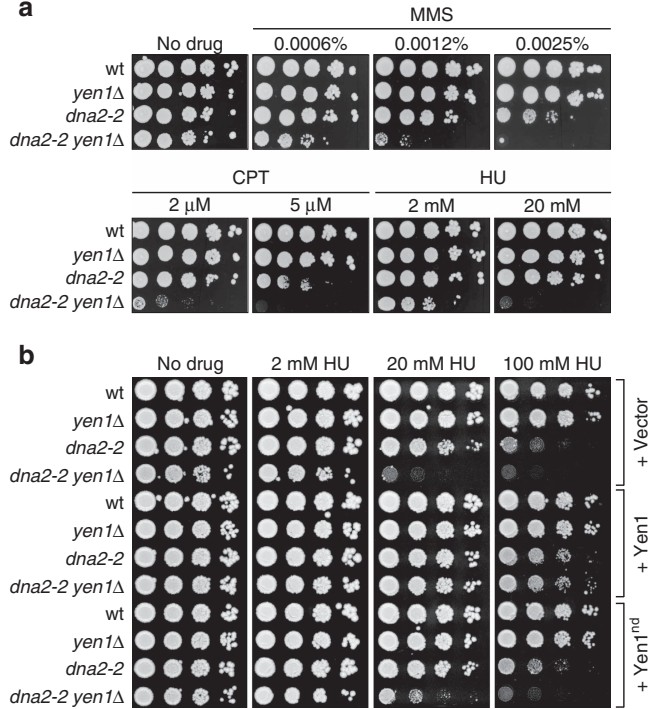

**Figure 3 | Dna2 helicase dysfunction and loss of Yen1 nuclease activity synergistically sensitize cells to exogenous replication stress.** (**a**) Drop assays to determine the drug-sensitivity of the indicated strains were done by spotting normalized tenfold serial dilutions of exponentially growing cells onto YPAD plates containing the indicated amounts of MMS, CPT or HU. (**b**) Analysis of the effects of Yen1 overexpression. Cells of the indicated strains were transformed with empty vector, or derivatives encoding wild-type or nuclease-deficient (Yen1[nd]) versions of Yen1, and plated on YPAD medium in the presence or absence of HU.

suggesting that the functional overlap of Dna2 and Yen1 relates to replication fork stalling. This also suggests that endogenous replication problems are responsible for the growth defects of *dna2-2* and *dna2-2 yen1Δ* cells in unperturbed conditions.

Plasmid-based expression of Yen1 suppressed the HU sensitivity phenotype of *dna2-2 yen1Δ* cells. This suppression was strictly dependent on the nuclease activity of Yen1, demonstrating that Yen1 protects *dna2-2* cells through nucleolytic cleavage of otherwise toxic DNA intermediates (Fig. 3b).

**DNA damage follows acute replication stress in *dna2-2* cells.** To investigate the immediate effects of replication stress on Dna2 helicase-defective cells, we next performed mitotic time-course experiments (Fig. 4). Synchronized *dna2-2* and *dna2-2 yen1Δ* cells were released into S phase under mild replication stress conditions in the presence of 50 mM HU, which impairs, but does not block, replication. After 2 h, cells were shifted back to drug-free medium. DNA synthesis was monitored by flow cytometry, while DNA replication/DNA damage checkpoint activation was assessed by Western blot analysis of the phosphorylation status of Rad53. As expected, wild-type and *yen1Δ* cells exhibited slowed replication progression in the presence of HU. Interestingly, *dna2-2* and *dna2-2 yen1Δ* cells progressed through S phase at a pace similar to wild-type. As shown in Fig. 4a, all strains showed S phase checkpoint activation in the presence of HU, and S phase checkpoint silencing occurred with normal kinetics across strains, followed by completion of

bulk DNA synthesis in drug-free medium. One hundred twenty minutes after removal of HU, wild-type and *yen1Δ* cells underwent cell division. In contrast, *dna2-2* and *dna2-2 yen1Δ* cells exhibited a reemergence of Rad53 phosphorylation and remained in G2/M with a 2 N DNA content. This biphasic Rad53 phosphorylation pattern, with an unexpected second wave of checkpoint activation in G2/M phase, required both the presence of the *dna2-2* allele and replication stress in the preceding S phase (that is, it was not discernible above background in control experiments without HU; Supplementary Fig. 3). One interpretation of these observations is that the Dna2 helicase is involved in an immediate response to replication fork stalling, preventing the emergence of DNA structures that signal DNA damage in G2/M.

The presence of Yen1 could not protect Dna2 helicase-defective cells from G2/M checkpoint activation and cell cycle arrest after acute replication stress. However, in the presence of Yen1, the G2/M arrest proved more transient. Thus, G1 cells with a 1 N DNA content started to appear 240 min after removal of HU, and continued to appear through overnight incubation in the *dna2-2* culture, while the *dna2-2 yen1Δ* strain produced very few G1 cells, as judged by flow cytometry (Fig. 4a). Microscopic analysis showed that within the *dna2-2 yen1Δ* culture ~11% of cells had segregated their nuclear DNA, while the *dna2-2* culture contained a significantly higher number of cells, ~32%, with segregated DNA after overnight incubation (Fig. 4b). This correlated with a significantly lower lethality scored for *dna2-2* mutants (~28% viability compared to wild-type) than for the *dna2-2 yen1Δ* double mutant (<1% viability compared to wild-type) (Fig. 4c). We conclude that Yen1 promotes mitotic exit with viable chromosome segregation in Dna2 helicase-defective cells recovering from acute replication stress.

**Yen1 resolves *dna2-2* post-replicative chromosome links.** A potential explanation for the apparent slow recovery of *dna2-2* cells from G2/M checkpoint arrest relates to recent work showing that Yen1 activity is cell cycle-regulated, with cyclin-dependent kinase (CDK)-mediated phosphorylation lowering its catalytic activity and inhibiting access to the nucleus in S and G2 phase[40]. Upon anaphase onset, Cdc14-dependent dephosphorylation activates Yen1 and allows the protein to accumulate inside the nucleus during mitosis. To test whether Yen1 cell cycle control is manifest in Dna2 helicase-defective cells recovering from acute replication stress, we expressed and monitored a functional (Supplementary Fig. 4a) version of Yen1, tagged with enhanced green fluorescent protein (EGFP), in *dna2-2* and wild-type cells. The expected bi-phasic checkpoint activation of *dna2-2* cells in response to acute HU treatment was recapitulated in the presence of Yen1-EGFP (Supplementary Fig. 4b), and the fusion protein exhibited the characteristic cell cycle-dependent subcellular localization reported for Yen1 (Fig. 5a)[40]. Importantly, when cells accumulated as large-budded G2/M cells after HU wash-out, a subset of cells was double-nucleated with a nuclear Yen1-EGFP signal, indicating that mitotic entry had occurred. This subset of cells was markedly larger in case of the wild-type strain after 2 h in drug-free medium (Fig. 5b). After 4 h in drug-free medium, the fraction of wild-type G2/M cells diminished as cells underwent mitosis. In contrast, *dna2-2* cells remained mostly in G2/M, with Yen1-EGFP in the cytoplasm and a single nucleus at the bud neck, as expected for DNA damage checkpoint-mediated pre-anaphase arrest. Thus, targeting of Yen1 to the nucleus through the actions of Cdc14 (ref. 40) remained largely blocked, showing that unscheduled DNA damage checkpoint signalling in *dna2-2* cells is associated with retention of Yen1 in the cytoplasm, and that this may

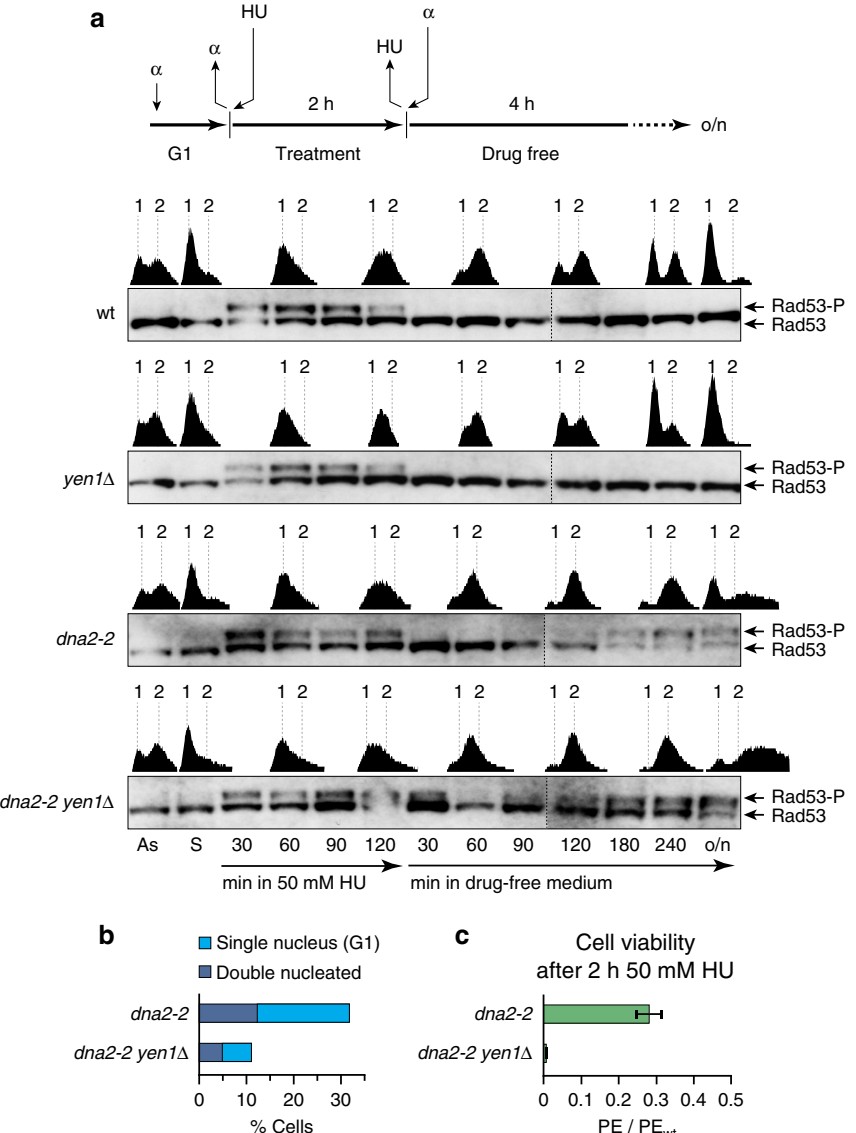

**Figure 4 | Dna2 helicase-defective cells are defective in the response to acute replication stress and require Yen1 for subsequent growth. (a)** Mitotic time-courses were performed as indicated. Cells, synchronized in G1, were released into acute replication stress in medium containing 50 mM HU for 2 h, followed by drug wash-out and incubation in drug-free medium with α-factor. Checkpoint activation and the progression of DNA replication were monitored by Western blot analysis of Rad53 phosphorylation (Rad53-P) and flow cytometry (1 and 2 N DNA content indicated). As, asynchronous; S, synchronous; o/n, overnight. **(b)** Quantification of single-nucleated G1 cells and double-nucleated G2/M cells in the indicated overnight yeast cultures shown in **a**. Two hundred cells per strain were analysed by microscopic inspection. **(c)** Viability of the indicated strains after acute replication stress. Cells were plated on drug-free YPAD medium and colony formation was quantified. The mean plating efficiency (PE) ± s.e.m. (n = 3) is presented relative to wild-type.

represent a major impediment to the recovery of Dna2 helicase-defective cells from replication stress.

To test this idea further, we next disrupted the G2/M DNA damage checkpoint by deletion of *RAD9*, allowing unrestrained anaphase entry, and thus Yen1 activation, in Dna2 helicase-defective cells. Upon acute replication stress treatment, checkpoint activation during S phase occurred normally in the absence of Rad9, consistent with signalling in response to replication fork stalling, rather than DNA damage, through the intact Mec1-Ddc2/Mrc1/Rad53-dependent pathway. In contrast, unscheduled Rad53 phosphorylation in G2/M phase after HU wash-out was abolished in *dna2-2 rad9Δ* and *dna2-2 yen1Δ rad9Δ* cells (Fig. 5c). Dna2 helicase-defective cells now progressed to cell division with kinetics similar to those exhibited by the *rad9Δ* control strain, and we were able to study the effects of Yen1 by microscopic inspection. This revealed two important

phenotypes associated with concomitant loss of Dna2 helicase function and Yen1. Sixty minutes after the removal of HU, *dna2-2 yen1Δ rad9Δ* samples contained roughly threefold higher levels of early anaphase cells characterized by an elongated nucleus stretched through the bud neck, as compared to *dna2-2 rad9Δ* and *rad9Δ* samples. Concomitantly, there was a delay in the appearance of G1 cells containing a single nucleus (Fig. 5d). This suggests that chromosome segregation and cytokinesis were physically impeded in the absence of Yen1. Consistently, and exclusively in *dna2-2 yen1Δ rad9Δ* cells, we observed prominent chromosomal DNA bridges that span the bud neck and connect the segregating masses of nuclear DNA (13.6 and 8% of the double-nucleated cells affected 180 and 240 min after HU wash-out, respectively) (Fig. 5e). In some instances this phenotype could be seen in cells approaching abscission, as indicated by a narrowing bud neck.

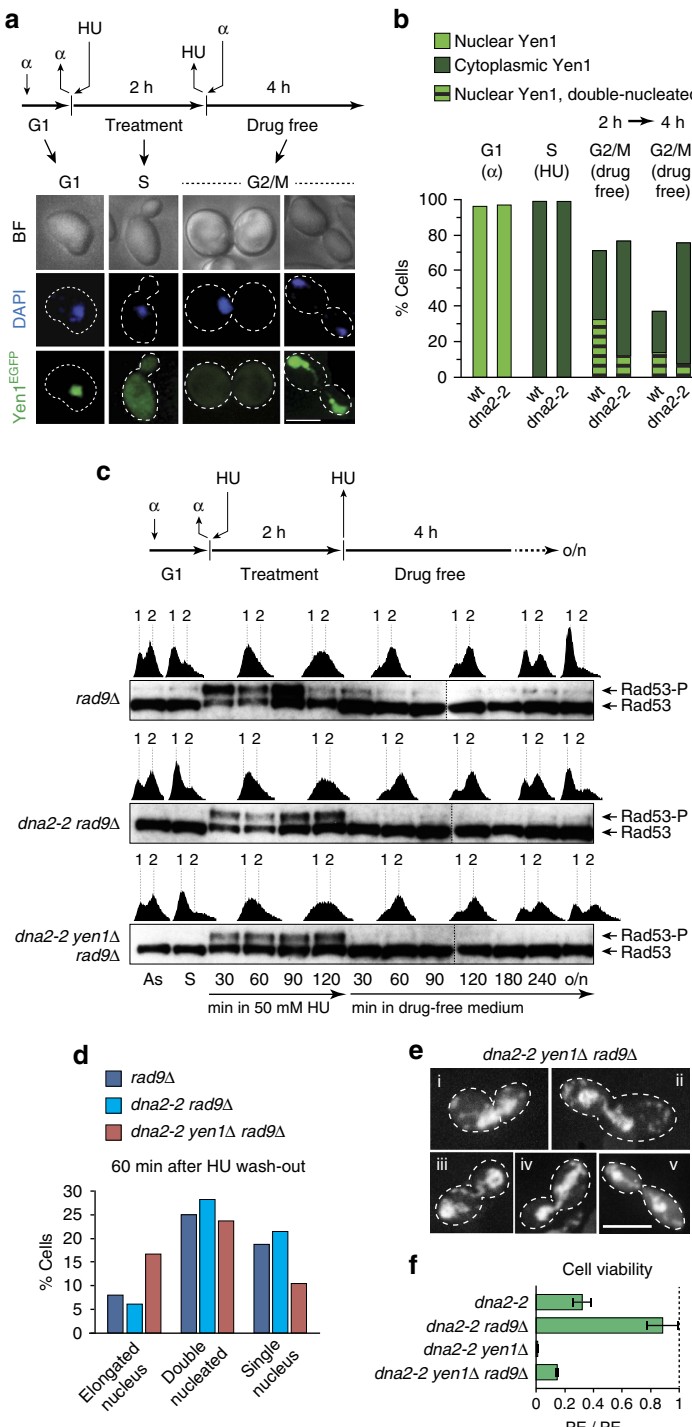

**Figure 5 | Post-replicative chromosomal links in Dna2 helicase-defective cells and mitotic resolution by Yen1.** (**a**) Subcellular localization of Yen1 in *dna2-2* cells exposed to acute replication stress. Mitotic time-courses were performed as indicated with wild-type and *dna2-2* cells expressing Yen1-EGFP. Samples were analysed for nuclear and cytoplasmic localization of Yen1-EGFP following α-factor arrest, 1 h after release into HU-containing medium, and 2 and 4 h after drug wash-out. Representative images are shown. Scale bar, 5 μm. (**b**) Quantitative view of Yen1-EGFP localization as determined in experiments such as those shown in **a** (≥100 cells scored for G1 and S phase per strain; ≥200 cells 2 and 4 h after drug wash-out). (**c**) Mitotic time-course experiment with DNA damage checkpoint-disrupted strains, performed as indicated and analysed as described for Fig. 4a. (**d**) Relative distribution of cells in early anaphase (elongated nucleus), late anaphase (double-nucleated) and post-cytokinesis (single-nucleated), as determined by microscopic inspection of samples from **c**, 60 min after HU wash-out (≥100 cells scored per strain). (**e**) Representative microscopic images showing an early anaphase cell with elongated nucleus spanning the bud neck (i), and late anaphase cells with chromosomal DNA bridges between the segregated masses of nuclear DNA (ii-v), a phenotype exclusively observed in *dna2-2 yen1Δ rad9Δ* cells. Cells treated as described for **c**. Scale bar, 5 μm. (**f**) Cell viability of the indicated strains, treated as in **c**, assessed by colony outgrowth. The mean plating efficiency (PE) ± s.e.m. (*n* = 3) is presented relative to *rad9Δ*.

When we determined the effect of checkpoint disruption on cell viability, we found that deletion of *RAD9* increased the viability of Dna2 helicase-defective cells after acute replication stress treatment threefold, reaching levels very similar to those observed for the *rad9Δ* control strain. In the absence of *YEN1*, viability was also improved, but did not reach more than ~14% of the viability of the *rad9Δ* control (Fig. 5f). Checkpoint activation in Dna2 helicase-defective cells after acute replication stress therefore appears futile, and eliminating the G2/M checkpoint enabled a highly effective Yen1-dependent survival pathway, while allowing a small subset of cells to survive in a Yen1-independent manner. As expected, checkpoint disruption had no beneficial effect when cells were exposed to chronic replication stress (Supplementary Fig. 5), consistent with improved survival being linked specifically to allowing Yen1 access to post-replicative lesions in Dna2 helicase-defective cells after acute replication stress. These observations resonate with previous findings showing that the lethality of some temperature-sensitive *dna2* alleles, and of *dna2Δ*, can be suppressed by deleting *RAD9* (refs 14,41,42), linking this phenomenon, at least for the Dna2 helicase-defective *dna2-2* allele, to the removal of the inhibitory effect of the G2/M DNA damage checkpoint on Yen1 activation.

To test whether Yen1 activation, not mitotic entry *per se*, is sufficient for Yen1 to resolve aberrant DNA intermediates that arise in Dna2 helicase-defective cells, we used a constitutively active form of Yen1, referred to as Yen1[on] (Supplementary Fig. 6a). Yen1[on] is not controlled by CDK and therefore permanently active and nuclear[43]. We expressed Yen1[on] from a galactose-inducible promoter in G2/M in *dna2-2 yen1Δ* cells recovering from acute, HU-induced replication stress in the presence of nocodazole. Strikingly, Yen1[on]-expressing cells did not exhibit unscheduled DNA damage checkpoint activation during nocodazole-induced G2/M arrest (Supplementary Fig. 6b). Furthermore, transient expression of Yen1[on] in *dna2-2* cells recovering from acute HU treatment led to a significant increase in cell viability (~3.3-fold ± 0.24 s.e.m., $n = 2$), as determined by colony outgrowth.

Collectively, these results suggest that Dna2 helicase-defective cells fail to respond adequately to replication stress, leading to post-replicative DNA damage signalling and chromosome entanglements. Upon anaphase entry, Yen1 promotes the survival of Dna2 helicase-defective cells by resolving post-replicative chromosomal DNA links, allowing proper chromosome segregation.

**Yen1 acts distinct from HJ resolution in *dna2-2* cells.** Yen1 is known for its role in removing persistent HJ DNA structures that accumulate as Rad52-dependent HR intermediates. Previous findings also suggest an increased requirement for Rad52-dependent DNA repair by HR in Dna2 helicase-defective cells. Thus, *dna2-2 rad52Δ* cells without overt growth defect at 30 °C, but temperature sensitivity at a restrictive temperature of 37 °C, have been described[44]. We generated *dna2-2 rad52Δ* cells and found that compared to *dna2-2* (103 min) and *rad52Δ* (112 min), double mutant cells grew slowly, even at 30 °C, with a doubling time of 144 min. Furthermore, loss of Rad52 led to pronounced synthetic hypersensitivity of *dna2-2* cells to HU (Fig. 6a). Finally, we observed a significant increase of spontaneous Rad52 foci, indicative of HR[45], in *dna2-2* cells in unperturbed conditions (Fig. 6b), whereas Dna2 focus formation is elevated in *rad52Δ* cells[46]. This suggests that Dna2 and Rad52-dependent HR represent parallel and compensatory pathways in the response to replication stress.

Elevated levels of HR repair could explain why loss of Yen1 is detrimental to *dna2-2* cells. If so, *dna2-2 mus81Δ* cells should exhibit an even stronger growth defect than *dna2-2 yen1Δ* cells, given that Mus81-Mms4 is activated, in a CDK-dependent manner, prior to Yen1 activation at anaphase onset. Reaching an activity peak in its hyperphosphorylated state in G2/M, Mus81-Mms4 is thus the major nuclease in removing HR intermediates in budding yeast[40]. Notwithstanding, we found that in contrast to loss of Yen1, disruption of Mus81-Mms4, or the Slx1-Slx4 HJ resolvase, did not increase the doubling time of *dna2-2* cells in unperturbed conditions. In the presence of HU or MMS, deletion of *SLX1* had no effect on the sensitivity of Dna2 helicase-defective cells. Deletion of *MUS81*, which in itself results in replication stress sensitivity, added to their sensitivity (Fig. 6c and data not shown), consistent with a requirement for Mus81-Mms4 in the resolution of excessive HR intermediates in *dna2-2* cells. However, *dna2-2 yen1Δ* cells were significantly more sensitive to HU or MMS than *dna2-2 mus81Δ* cells (0.2 versus 45% cell survival at 20 mM HU as determined by colony outgrowth) (Fig. 6c,d and data not shown), despite the fact that defects related to HJ resolution as a consequence of Yen1 loss have been shown to transpire only in the absence of a functional Mus81-Mms4 resolvase[39]. Therefore, there is a pathway, distinct from canonical HJ resolution, which uniquely requires Yen1 for the removal of DNA intermediates that are apparently not amenable to cleavage by Mus81-Mms4, in *dna2-2* cells. Indeed, the toxicity caused by loss of Yen1 cannot be explained by an accumulation of HR intermediates alone, as we found that the synthetic sick relationship between *DNA2* and *YEN1* is maintained in cells deleted for *RAD52*, which cannot engage in HR reactions. In fact, we were unable to generate a *dna2-2 rad52Δ yen1Δ* triple mutant by tetrad dissection (data not shown), and have confirmed an essential requirement for Yen1 in *dna2-2 rad52Δ* cells using a plasmid-based assay (Fig. 6e). These results contrast with an epistatic relationship that exists between *RAD52* and the HJ resolution pathway defined by *MUS81-MMS4* and *YEN1* (refs 39,47), and imply that the structures that are targeted by Yen1 in order to maintain the viability of Dna2 helicase-defective cells derive from perturbed replication intermediates in a HR-independent manner.

**Yen1 resolves DNA replication intermediates in *dna2-2* cells.** To address the question whether replication fork stalling in Dna2 helicase-defective cells gives rise to an accumulation of DNA intermediates that might become targets for Yen1, we turned to the natural replication fork barrier (RFB)[48] within the ribosomal DNA (rDNA) on chromosome XII. We compared rDNA from actively replicating wild-type and *dna2-2* mutant cells by two-dimensional (2D) gel electrophoresis and monitored the disappearance of replication intermediates as cells progressed from S phase to nocodazole-induced G2/M arrest. In S phase, Dna2 helicase-defective cells showed a pattern of replication intermediates very similar to wild-type (Fig. 6f). As expected, replication intermediate levels dropped significantly when cells accumulated in G2/M during nocodazole arrest. However, the resolution of replication intermediates, in particular of RFB-stalled and converged forks, was less efficient in *dna2-2* cells, leading to a ~2 and ~3-fold less prominent decrease compared to wild-type, respectively. These results are in good agreement with previous observations of accumulating stalled and converged fork intermediates within the rDNA of *dna2-2* cells[49], and corroborate the notion of an aberrant response to replication fork stalling in Dna2 helicase-defective cells. To see if Yen1 targets aberrant replication intermediates that persist in Dna2 helicase-defective cells, we expressed, in asynchronous *dna2-2* cultures, constitutively active Yen1[on] (Fig. 6g), allowing us to monitor Yen1 actions prior to the activation of endogenous Yen1 and chromosome segregation in M phase. Similar to staged S phase cells, 2D gel electrophoresis of exponentially growing

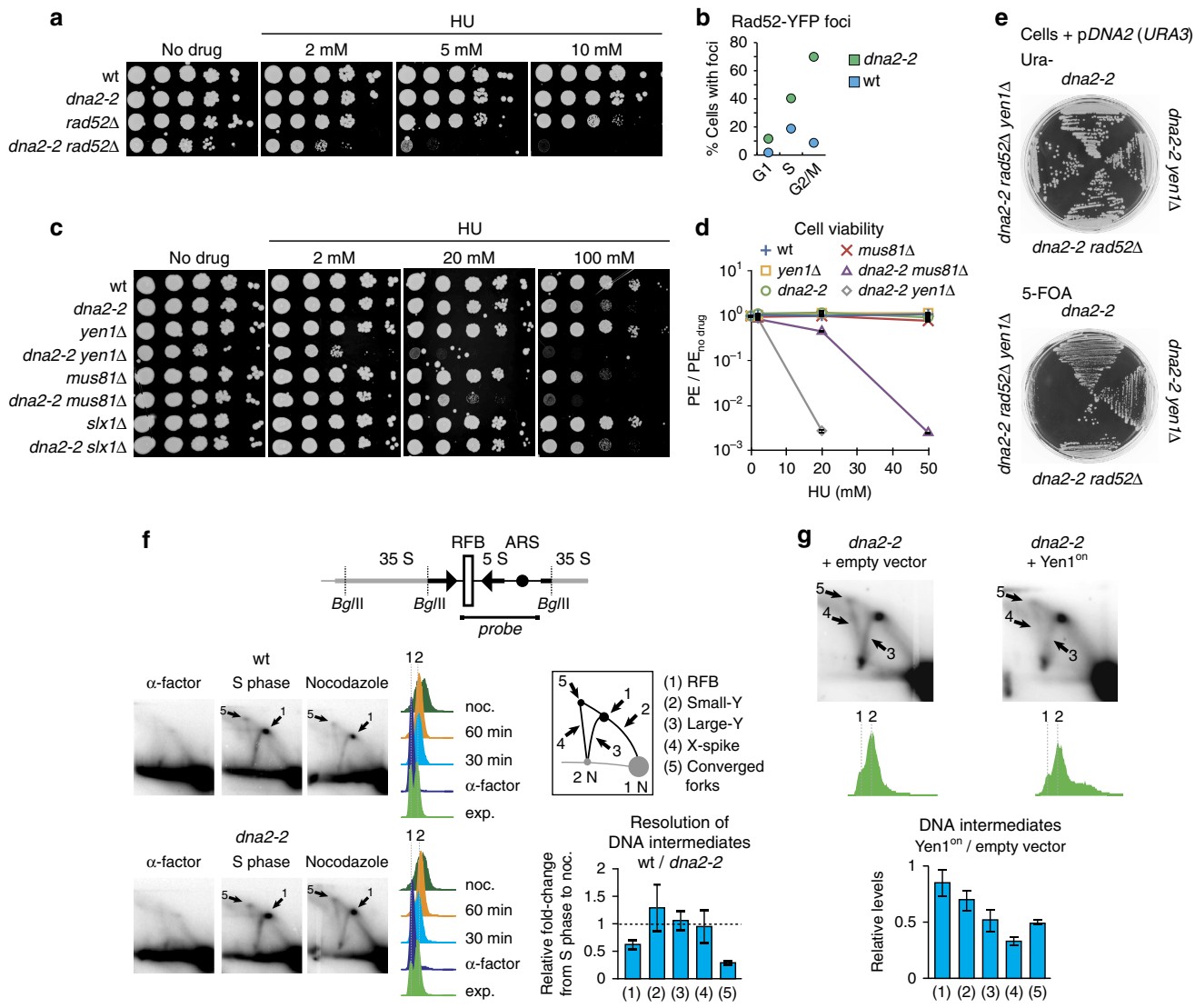

**Figure 6 | Yen1 uniquely resolves toxic DNA intermediates in Dna2 helicase-defective cells along a pathway distinct from canonical HJ resolution.**
(**a**) Synergistic defects in the resistance to replication stress in homologous recombination-deficient *dna2-2 rad52Δ* cells. Drop assay performed as in Fig. 3a. (**b**) Spontaneous Rad52-YFP foci in wild-type and *dna2-2* cells in different cell cycle stages, determined by microscopic analysis (≥180 cells scored per strain). (**c**) Genetic interactions of *dna2-2* with HJ resolvases *YEN1*, *MUS81-MMS4* and *SLX1-SLX4*. Drop assays performed as in Fig. 3a. (**d**) Mean plating efficiency (PE) ± s.e.m. (*n* = 3) of the indicated strains assessed by colony outgrowth in the presence of increasing amounts of HU, relative to no-drug conditions. (**e**) Interaction between *dna2-2*, *RAD52* and *YEN1*. Cells of the indicated genotypes and containing a plasmid expressing wild-type *DNA2* (p*DNA2*) were grown under uracil selection to ensure retention of the p*DNA2*, or on medium containing 5-FOA to select against the plasmid. Failure to grow on 5-FOA is indicative of an inviable genotype. (**f**) Analysis of replication and recombination intermediates in the rDNA of wild-type and *dna2-2* cells traversing S phase into nocodazole-induced G2/M arrest. Genomic DNA was digested with *Bgl*II and subjected to 2D gel electrophoresis. The fragment probed by Southern hybridization contained the rDNA autonomously replicating sequence (ARS), the 5S transcriptional unit, and the RFB in its centre, as indicated. DNA structures chosen for quantification included RFB-arrested forks (1), Y-arc structures containing a replication fork at varying positions outside the RFB (2,3), the X-spike indicative of four-way branched DNAs containing Holliday junctions or hemicatenanes (4), and forks converging at the RFB (5). Representative autoradiographies are marked for RFB-stalled (1) and converged replication fork intermediates (5), which were resolved less efficiently in *dna2-2* cells compared to wild-type following S phase. Three independent experiments were quantified and the data are presented as mean values ± s.e.m. (**g**) 2 D analysis as in **f**, but using exponentially growing *dna2-2* cells expressing or not Yen1$^{on}$. Intermediates that were reduced upon Yen1$^{on}$ expression are indicated. Three independent experiments were quantified and the data are presented as mean values ± s.e.m.

*dna2-2* cells showed the expected rDNA replication intermediates and, in addition, a more prominent signal indicative of recombination intermediates (X-spike), consistent with an accumulation of G2/M cells with increased rates of HR[49]. Yen1$^{on}$ did not affect the RFB signal, suggesting that replication forks arrested at the barrier are not immediately susceptible to Yen1 nuclease activity. In contrast, and consistent with the ability of Yen1 to resolve recombination intermediates, moderate constitutive Yen1$^{on}$ expression markedly reduced the X-spike signal. Importantly, single fork intermediates (Y structures) and converged forks were also decreased on Yen1$^{on}$ expression, showing that Yen1, in addition to resolving four-way X-DNA, is able to remove replication intermediates that accumulate in Dna2 helicase-defective cells. These results suggest that endogenous Yen1, activated on anaphase entry, uniquely resolves persistent replication fork/converging fork structures to disentangle

underreplicated nascent sister chromatids when Dna2 helicase-defective cells enter mitosis, thereby safeguarding chromosome segregation and enabling viable mitotic exit.

## Discussion

Our analyses of the interplay between Dna2, HJ resolvase Yen1 and the DNA damage checkpoint allows us to define important functions of the Dna2 helicase activity and the Yen1 nuclease in the replication stress response. We propose a model, where the Dna2 helicase activity represents a HR-independent replication stress response pathway that helps to ensure full replication of the genome. Replication intermediates that escape the attention of Dna2 persist and impair sister chromatid separation, unless they are resolved by Yen1 in mitosis. Thus, the actions of Yen1, which has so far only been known to target HR intermediates, allow viable chromosome segregation along a novel pathway, distinct from canonical HJ resolution (Fig. 7).

The precise constitution of the DNA structures that threaten chromosome segregation in Dna2 helicase-defective cells remains to be determined. However, the fact that Yen1 can detoxify them indicates that these DNA intermediates conform to the substrate spectrum of Yen1, which includes 5′-flaps and fully double-stranded DNA three-way and four-way junctions[29,43]. As Dna2[R1253Q] retains 5′-flap endonuclease activity (Fig. 1f), this activity of Yen1, which could potentially support the proposed role of Dna2 in Okazaki fragment processing[18,19], is unlikely to explain the requirement for Yen1 in Dna2 helicase-defective cells. Consistently, we find no genetic indication that Yen1 might support functions of the major Okazaki fragment processing nuclease Rad27 in vivo (Supplementary Fig. 7). We thus favour the possibility that the capacity of Yen1 to target branched dsDNA intermediates is relevant for the protection of dna2-2 cells. This ability distinguishes Yen1, Mus81-Mms4 and Slx1-Slx4 from other structure-specific nucleases, and allows them to resolve HR-dependent HJs, but also analogues of replication forks, and presumably reversed fork intermediates, which are structurally equivalent to four-way HJs[30,32,50]. DNA intermediates that require detoxification by Yen1 arise in Dna2 helicase-defective cells in the absence of Rad52 (Fig. 6e), and in the presence of Mus81-Mms4, which we find in the active, hyperphosphorylated[40] form in post-replicative dna2-2 cells (Supplementary Fig. 8). Therefore, Yen1 appears not to be primarily required to remove HR intermediates in dna2-2 cells, but instead for removing persistent replication intermediates, such as arrested forks or converged forks that fail to fuse (Fig. 6f,g). Of note, a similar activity of MUS81-EME1 towards late replication intermediates has been shown to avoid sister chromatid non-disjunction in human cells[33,34]. Thus, Yen1 and other HJ resolving enzymes might function in complementary fashion, rather than redundantly, in targeting dead-end replication intermediates to protect chromosome segregation.

Persistent replication intermediates could explain chromosome non-disjunction in Dna2 helicase-defective cells (Fig. 5e), and it is tempting to speculate that the Dna2 helicase activity may be involved in replication fork remodelling reactions that facilitate fork recovery. This would be conceptually similar to the role of the Dna2 nuclease in replication restart at reversed forks through degradation of the regressed DNA branch[21–23]. A particularly attractive possibility is that fork reversal might occur and/or persist as a consequence of Dna2 helicase dysfunction. The resulting chicken-foot structure, which effectively contains a single-ended DNA double-strand break at the tip of the regressed DNA branch, could account for DNA damage checkpoint activation[51], which we observe in Dna2 helicase-defective cells (Figs 2d,4a and 5c). A four-way chicken-foot DNA intermediate would also be amenable to resolution by Yen1; perhaps more so than by Mus81-Mms4, which is greatly stimulated by pre-existing nicks within DNA junctions, such as those that may be present in HR-dependent joint molecules and maturing HJs prior to a final ligation step[29,30,43]. Importantly, and regardless of their exact structural features, the intermediates resolved by Yen1 in Dna2 helicase-defective cells constitute a first DNA target that is uniquely processed by Yen1. This demonstrates greater complexity in the uses of HJ resolvases in cells, and could explain the evolutionary conservation of Yen1/GEN1.

Intriguingly, Dna2 and Yen1 are both subject to CDK1-regulated nucleocytoplasmic shuttling[52] (Fig. 7b). During S phase, phosphorylation of Yen1 mediates nuclear exclusion, whereas phospho-Dna2 accumulates inside the nucleus. Thus, Dna2 can access sites of impaired DNA replication, and, consistently, has been found to form discrete nuclear foci during HU-induced replication arrest[53]. Dna2 helicase dysfunction gives rise to lesions that require Yen1 for resolution, but also triggers G2/M checkpoint activation, precluding dephosphorylation-dependent Yen1 activation and translocation to the nucleus on anaphase entry (Fig. 5a). Paradoxical though it may seem, this likely reflects a trade-off between the need to protect chromosomes from the DNA de-branching activities of Yen1 during S phase, while exploiting its unique biochemical properties to remove persistent chromosomal DNA links in M phase. Indeed, Yen1[on] expression in G2/M allows resolution of toxic DNA intermediates in dna2-2 cells (Supplementary Fig. 6), but constitutive expression is associated with MMS sensitivity, and tight control over the activities of HJ resolvases has been shown to limit sister chromatid exchange and the risk of loss of heterozygosity[40,54–56]. Despite the risk of terminal G2/M arrest, Yen1 effectively maintains the viability of Dna2 helicase-defective cells in unperturbed conditions (Fig. 2a), indicating its late activation as an elegant failsafe mechanism that allows Yen1 to act indiscriminately on DNA structures that resemble normal replication intermediates, identified as aberrant only by their presence at the wrong time in the cell cycle. In future, it will be interesting to see whether Yen1 represents a more general surveillance nuclease for aberrant replication intermediates that persist into anaphase. Strong G2/M checkpoint signalling and terminal arrest at the G2/M boundary might have precluded the detection of Yen1 functions downstream of repair and replication factors other than DNA2 in large-scale screening efforts thus far.

Loss and overexpression of DNA2 has been observed in human cancers and cancer cell lines[12,57,58], while haploinsufficiency promotes cancer formation in heterozygous DNA2-knockout mice[9]. This suggests a complex role in cancer, where genome instability caused by impaired DNA2 function may drive tumorigenesis, whereas upregulation of DNA2 may help cancer cells to survive continuous DNA replication stress. Interestingly, a homozygous mutation in DNA2 has recently been identified in patients with Seckel syndrome[59], a disease associated with a compromised response to replication fork stalling on the cellular level[60]. Depletion of DNA2 in mammalian cells recapitulates many of the phenotypes seen in Dna2-defective yeast, including sensitivity to replication stress, elevated DNA damage, chromosome instability and G2/M cell cycle delay[9–12], indicating functional conservation. Our results implicate the elusive Dna2 helicase in replication fork recovery. Yen1 provides a downstream survival pathway, along which toxic DNA intermediates that arise when the Dna2 helicase activity fails to respond adequately to replication fork stalling are resolved. Similar two-tiered mechanisms may contribute to the aetiology of human pathologies involving DNA2.

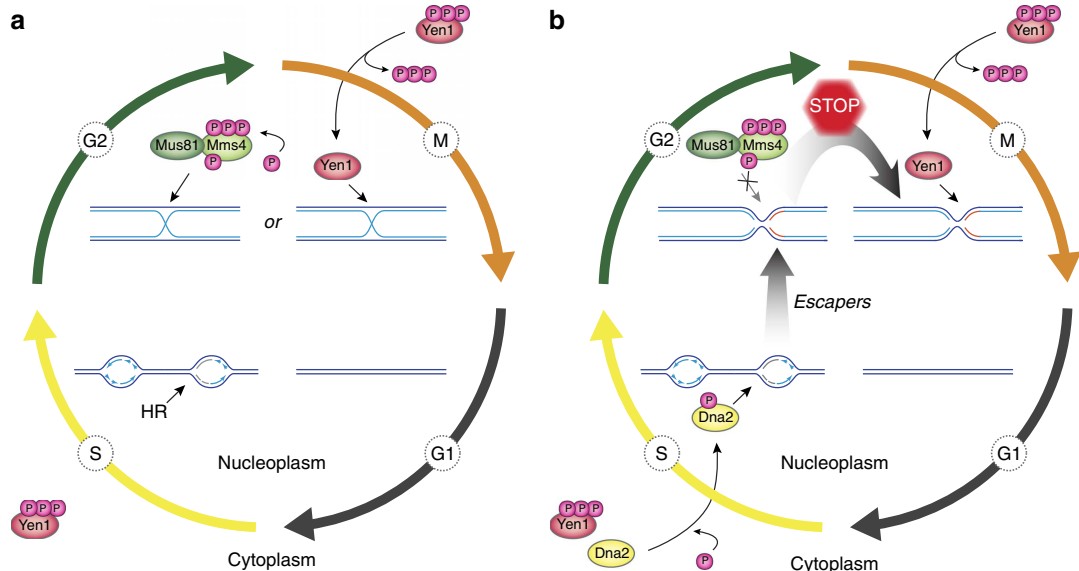

**Figure 7 | Model for a two-tiered response to replication stress by Dna2 and Yen1.** (**a**) Canonical role of Yen1 in the resolution of HR intermediates that arise during HR-mediated recovery of stalled replication forks. Mus81-Mms4 is activated by hyperphosphorylation in G2/M, prior to activation and nuclear import of Yen1 upon anaphase onset. Thus, HR intermediates are predominantly cleaved by Mus81-Mms4, with Yen1 acting as a catchall in M phase to remove persistent recombination structures in time for chromosome segregation. (**b**) Parallel to HR, the Dna2 helicase is tending to stalled replication forks. Replication intermediates that escape the attention of Dna2 give rise to toxic structures that are sensed by the DNA damage checkpoint, but which are refractory to processing by Mus81-Mms4. At anaphase entry, Yen1 is activated and uniquely resolves persistent replication intermediates, averting mitotic catastrophe. See text for details.

## Methods

**Recombinant proteins.** Wild-type Dna2, Dna2$^{R1253Q}$, Dna2$^{E675A}$ and Dna2$^{K1080E}$ were expressed from a modified pGAL:DNA2 vector, adding N-terminal FLAG and HA tags and a C-terminal 6 × His tag, and purified as described previously[36,37]. RPA protein was expressed and purified as described[61].

**Nuclease/helicase and ATPase assays.** Experiments were carried out and analysed as described[36,37]. In brief, we used an assay that couples ATP hydrolysis to oxidation of NADH to determine the rate of ATP hydrolysis by Dna2 variants by following the decrease in optical absorbance by NADH at 340 nm over time. Kinetic plots of ATP hydrolysis were derived by calculating the amount of ATP hydrolysed per time interval. Fifteen microlitres reactions contained 25 mM Tris-acetate (pH 7.5), 2 mM magnesium acetate, 1 mM ATP, 1 mM dithiothreitol, 0.1 mg ml$^{-1}$ bovine serum albumin, 1 mM phosphoenolpyruvate, 16 U ml$^{-1}$ pyruvate kinase, 1 nM DNA substrate, 16.8 nM RPA and Dna2 proteins as indicated. Nuclease assays were incubated at 30 °C for 30 min. For analysis by denaturing polyacrylamide electrophoresis, samples were heat-denatured in formamide. DNA substrates were assembled using oligonucleotides X12-3 and X12-4SC, and PC 92 and X12-4SC for the 19 and 30 nt 5′-tailed DNA substrates, respectively[36,37]; the 5′-flapped DNA substrate consisted of oligonucleotides X12-4NC, Flap 19 X12-4C, and 292, as described[37]. Where indicated, oligonucleotides were $^{32}$P-labelled at the 5′-end using [γ-$^{32}$P] ATP and T4 polynucleotide kinase (New England Biolabs). Unincorporated nucleotides were removed using MicroSpin G25 columns (GE Healthcare) before annealing the respective DNA substrates.

**Yeast strains and plasmids.** S. cerevisiae strains (Supplementary Table 1) were derived from BY4741 (ref. 62) using standard methods. The dna2-2 allele was generated using pop-in/pop-out mutagenesis[63], and the DNA damage sensitivity of the resulting strain could be complemented with plasmid-borne wild-type DNA2 cloned into vector pAG416GPD-ccdB (pDNA2) (Supplementary Fig. 2b). For constitutive expression, YEN1 was cloned into vector pAG416GPD-ccdB or pAG416GPD-ccdB-EGFP[64], and site-directed mutagenesis was performed to generate a catalytically inactive form of Yen1 bearing the mutations E193A and E195A. Yellow fluorescent protein (YFP)-tagged Rad52 was expressed from its endogenous promoter using centromeric plasmid pWJ1213 (ref. 65). If not stated otherwise, all strains were cultured at 30 °C using YPAD media. YEN1$^{on}$ was cloned into vector pAG416GPD-ccdB, or pYES-DEST52 (Invitrogen) for expression from a GAL1 promoter in YPLG medium with 2% (w/v) galactose and 1% (w/v) raffinose. Antibodies used to monitor the expression of tagged proteins were Abcam mouse monoclonal anti-V5 antibody ab27671, and Sigma mouse monoclonal anti-Myc antibody 9E10. Santa Cruz Biotechnology goat polyclonal

anti-Mcm2 antibody yN-19 was routinely used to ensure gel lanes were equally loaded for total protein.

**Cell viability and drug sensitivity assays.** Doubling times were determined as described[66] and averaged over at least three independent experiments. For microscopic determination of cell cycle stage (budding index), an average of 400 cells per strain and replicate were scored. Plating efficiency as a measure of strain viability was determined by colony outgrowth after plating a defined number of cells. The number of colonies formed after 3–4 days at 30 °C was divided by the number of cells plated as quantified in haemocytometer counts. For drop assays, exponentially growing cells were normalized to 10$^7$ cells ml$^{-1}$, and 2 μl drops of tenfold serial dilutions were spotted onto the appropriate medium with or without MMS, HU or CPT. If not stated otherwise, plates were incubated for 3–4 days at 30 °C. For liquid survival assays, overnight cultures were diluted to OD$_{600}$ = 0.1–0.2 and grown for 4 h, then synchronized with α-factor in G1 and released into YPAD containing 50 mM HU for 120 min. Relevant dilutions were plated onto YPAD plates and colonies were counted after 3–4 days.

**Mitotic time-courses.** For time-course experiments, cells were grown exponentially (OD$_{600}$ = 0.4–0.6) and synchronized by addition of α-factor (routinely > 95% unbudded cells for wild-type, ≥ 90% for Dna2 helicase-defective strains). Cells were then harvested, washed and released into YPAD containing 50 mM HU for 2 h. After HU wash-out, cells were cultured in drug-free medium. Aliquots for flow cytometry, Western blot analysis and microscopy were withdrawn at regular intervals. Where indicated, α-factor or nocodazole (15 μg ml$^{-1}$) was added during and/or after treatment.

**Analysis of Rad53-phosphorylation.** TCA-precipitated proteins were separated by SDS-PAGE using precast gels (Invitrogen), and blotted onto polyvinylidene difluoride membranes using a Bio-Rad Turbo blot system. Rad53 protein was detected using a custom-made mouse monoclonal antibody[67]. Uncropped immunoblots are shown in Supplementary Fig. 9.

**Flow cytometry.** Cells were fixed overnight in 70% ethanol at 4 °C with rotation and processed as described[68]. Cells were then washed and resuspended using 50 mM Na-citrate (pH 7). After brief sonication, RNase A was added (0.25 mg ml$^{-1}$), and cells were incubated overnight at 37 °C, washed, and resuspended in 50 mM Na-citrate (pH 7) containing 16 μg ml$^{-1}$ propidium iodide. Measurements of DNA content were done using a BD LSR II flow cytometer

(Becton Dickinson) operated with BD FACSDiva software. Data was processed with FlowJo (TreeStar).

**Microscopy.** DIC images were obtained using a Zeiss Axio Imager Z1 with a Plan-Apochromat 63 ×/1.4 DIC oil objective (Zeiss) and an AxioCam camera controlled by ZEN Blue 2012 software. To analyse nuclear DNA and chromosome segregation, cells were fixed with 70% ethanol for 5 min at room temperature and stained with 4,6-diamidino-2-phenylindole (DAPI) (50 ng ml$^{-1}$). For Yen1-EGFP and Rad52-YFP analyses, cells were fixed in 4% paraformaldehyde for 3 min at room temperature and stained with DAPI. Confocal images were collected using a Zeiss Axio Imager M1/Yokogawa CSU-X1 scanhead multipoint confocal microscope with a Plan-Neofluar 100 ×/1.45 oil objective and EM-CCD Cascade II camera (Photometrics) controlled by Metamorph 7.7.2 software (Molecular Devices), or a Rolera Thunder Back Illuminated EM-CCD camera (Q Imaging) controlled by VisiView software (Visitron Systems). Stacks of > 20 optical slices separated by 200 nm were collected, and images of two-dimensional projections were prepared with ImageJ software (Fiji).

**Analysis of rDNA replication by 2D gel electrophoresis.** Synchronized or exponentially growing cells were harvested by centrifugation, and genomic DNA was purified using G-20 columns (Qiagen) before digestion with *Bgl*II. For S phase samples, aliquots were withdrawn every 10 min for 60 min on release from α-factor-induced G1 arrest, and pooled before preparing genomic DNA. Ethanol-purified DNA digests (2.5 μg) were subjected to 2D gel analysis as described[69], with minor modifications. The first dimension gel (0.4% agarose in TBE) was run at 1 V cm$^{-1}$ at room temperature for 16 h. The second dimension gel (1.5% in TBE containing 0.3 g ml$^{-1}$ ethidium bromide) was run at 5 V cm$^{-1}$ at 4 °C in circulating TBE buffer for 5 h. The DNA was then blotted onto Hybond XL membrane (Amersham) by capillary transfer in 0.4 N NaOH. After UV-crosslinking, the membrane was blocked with ssDNA, probed for rDNA, and washed according to instructions by the manufacturer. A DNA template for the Southern probe was prepared by PCR from genomic DNA using primers 5′-GCCATTTACAAAAACATAACG and 5′-GGGCCTAGTTTAGAGAGAAGT[49]. The radiolabelled probe was then synthesized in the presence of [α-$^{32}$P] dCTP and [α-$^{32}$P] dATP using Klenow fragment polymerization (New England Biolabs/Bioconcept). Radioactive Southern blots were imaged using a phosphorimager screen (Kodak) and a Typhoon$^{TM}$ 9400 system (GE Healthcare), and quantified using ImageQuant TL v2005 software as described[70]. In brief, the signal intensities for individual image objects were normalized to the intensity of the 1 N spot after background correction. The fold change was calculated by dividing the normalized signal intensity of each intermediate in G2 phase by the corresponding signal in S phase. For the experiment with Yen1$^{on}$ expression in *dna2-2* cells, the normalized signal intensity for each scrutinized DNA intermediate in the strain harbouring the Yen1$^{on}$ construct was divided by the corresponding signal in the strain with empty vector.

**Data availability.** The authors declare that all data supporting the findings of this study are available within the article and its Supplementary Information, or from the corresponding author on request.

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

## Acknowledgements

We are indebted to S. Gasser, K. Shimada and A. Seeber for their generous technical help and reagents. We thank our colleagues S. Gasser, N. Thomä, and members of the Gasser and Rass laboratories for critical reading of the manuscript and helpful discussions. S. West and M. Blanco kindly provided the Yen1on construct. We are grateful for the assistance of H. Kohler from the Cell Sorting platform, and L. Gelman and S. Bourke from the Facility for Advanced Imaging and Microscopy at the Friedrich Miescher Institute. Work in the laboratory of P.C. is supported by Swiss National Science Foundation Professorship PP00P3 133636, and work in the laboratory of U.R. by the Swiss Cancer League & Swiss Cancer Research and the Novartis Research Foundation.

## Author contributions

G.Ö. and U.R. planned and analysed the experiments. G.Ö. performed the experiments with help from D.K. B.F. supported the two-dimensional gel analyses, and G.A.F. imaging and microscopy. M.L. and P.C. purified and analysed Dna2 *in vitro*. U.R. wrote the paper.

## Additional information

**Competing financial interests:** The authors declare no competing financial interests.

**How to cite this article:** Ölmezer, G. *et al.* Replication intermediates that escape Dna2 activity are processed by Holliday junction resolvase Yen1. *Nat. Commun.* **7**, 13157 doi: 10.1038/ncomms13157 (2016).

