## [Peer Review File · Nature Communications]

Reviewers' comments:

Reviewer #1 (Remarks to the Author):

1. This manuscript claims to show that DNA2 helicase deficiency leads to aberrant replication structures that persist into mitosis if not repaired by YEN1. However, the data at this stage are insufficient to support the claim. First, the authors never show that DNA2 deficiency leads to aberrant intermediates, which they propose are X-shaped structures involving sister chromatid junctions (resembling YEN1 substrates) in mitosis. Fig. 1e, the PFGE experiment, is completely unconvincing, is not quantified, and in any case is not state of the art. The authors need to carry out 2D gels of replication intermediates at a defined genomic locus, a much more precise probe of incomplete DNA replication than HU treatment (see study of Sgs1 by Foiani, Branzei and Rothstein and colleagues, Bernstein et al., 2009). In the absence of Dna2, these gels should show an increase in species that have been shown correspond to various interlinked, unresolved sister chromatid forms and these should increase in the *dna2 yen1* mutant (see Gianatassio et al. 2014 NSMB).

2. Second, they claim to show that it is the DNA2 helicase that is responsible for resolving such putative sister chromatid links, and that no role for the DNA2 helicase has been shown before. Ignoring for the moment decades of biochemistry that show that there is crosstalk between the helicase and nuclease, probably through shared substrate binding sites, they use only a single mutant, *dna2-2*. Contrary to the author's claim, it is not known if *dna2-2* protein is deficient in nuclease. Although *dna2-2* maps in the helicase domain, it maps in a motif involved in other helicases in DNA binding, which may affect both helicase and nuclease. The *dna2-2* protein has never been purified, so it has never been shown to have intact nuclease. The authors cite Zhu et al to support their claim, but that reference is insufficient since Zhu et al. state that *dna2-2* was shown to have intact nuclease by three references. However, none of these papers reports on purification and assay of *dna2-2*. Furthermore, Zhu et al does not show that *dna2-2* protein at normal levels can support resection, since they only show that overproducing *dna2-2* restores resection in *dna2 pif1* delete. Even overexpression of *dna2-1*, a temperature sensitive mutant and which is deficient in both helicase and nuclease, suppresses *dna2-1* phenotypes. You simply need more enzyme in these hypomorphs. The authors must express and purify *dna2-2* and show that nuclease is intact if they are to keep this claim that the helicase is behind the defects observed. Alternatively, they should use a mutant defective only in helicase, such as K1080A. Alternatively drop the claim about the helicase and just say Dna2.

3. Third, the YEN1-On experiment is interesting and an interesting interpretation is given. However, it is not the only interpretation. The YEN1 might suppress defects in Okazaki fragment processing when it is

expressed in S phase, just as FEN1 and EXO1 overexpression suppress dna2 mutations. The authors should test if YEN1-On also suppresses a rad27 mutant to see if YEN1 can also function in OFP. Even if they do not find suppression, they cannot rule out a role in OFP. This is why it is crucial to actually demonstrate that junctional replication intermediates accumulate and are the culprit in dna2 and in dna2 yen1 double mutants (point 1).

4. Suppression by rad9 delete is complicated. It has also been shown twice before, and these papers, one of which may not have been published when this one was submitted, are not cited. Rad9 is an inhibitor of resection and must be removed by Fun30, which complicates their model.

Reviewer #2 (Remarks to the Author):

In their manuscript entitled "Dna2 nuclease helicase and Holliday junction resolvase Yen1 provide a two-tiered response to replication stress", Ölmezer and colleagues investigate the interplay between Dna2 and Yen1 and uncover an unexpected contribution of the Yen1 resolvase in dealing with a specific type of DNA damage in order to ensure genome stability.

Dna2 is an essential DNA nuclease helicase involved in pleiotropic cell vital processes ranging from Okazaki fragments maturation to DNA repair. The gene coding for Yen1, a non-essential structure specific nuclease targeting Holliday junctions in late G2/M phase of the cell cycle, was identified as a synthetic lethal interacting gene of the dna2-helicase deficient (dna2-2) allele.

In this study, cells constitutively expressing the helicase deficient allele dna2-2, present an increased G2/M-arrested subpopulation. Deletion of yen1 increases this G2/M subpopulation (as well as the doubling time), but more importantly, decreases the viability of dna2-2 cells. The authors raise here the idea that dna2-2 cells require Yen1 to resolve persistent DNA links in dna2-2 cells and undergo mitosis.

dna2-2 and dna2-2 yen1Δ cells undergoing recovery after replication stress (50mM HU allowing some dna2-2 cells to undergo mitosis after recovery) display similar replication profiles as well as identical Rad53 activation kinetics, further reinforcing the model in which dna2-2 cells suffer late replication problems rather than S-phase defects. Furthermore, this assay clearly shows a checkpoint reactivation (measured by Rad53 phosphorylation) in dna2-2 and dna2-2 yen1Δ at the time mitosis. Strikingly, deletion of rad9 suppresses dna2-2 checkpoint reactivation and restores the viability of dna2-2 after replicative stress in a yen1 dependent fashion. Here the authors demonstrate that dna2-2 G2/M arrest stands more as an impediment for Yen1 activity. This hypothesis got further weight in the recovery from

replicative stressed followed by nocodazole arrest, where the authors unveiled that Yen1 activation per se is required to resolve DNA intermediates accumulating in dna2-2 cell.

This sheds new lights in the function of Yen1 in a peculiar replicative stress condition (mediated by the dna2 helicase-deficient allele, dna2-2) and will certainly be of interest for the readership of Nature Communications. Nevertheless, in its actual shape, the manuscript suffers of some overstatements and would be probably gain in strength with further experimental evidences as well as with some clarifications.

Major concerns:

1. The key point of this work is the ability of Yen1 to resolve secondary DNA structures arising in cells expressing dna2 helicase deficient allele (dna2-2) following or not replication stress (HU treatment). Despite being highly reproducible, the PFGE quantitation (Fig1.e) is only an indirect evidence for the presence of post-replicative DNA inter-sister chromatids, and probably 2D gels analyses would be more informative. Quantitation of these intermediates in replicative stress conditions, in presence or absence of Yen1 could certainly strengthen the message.

2. Does the faint DNA signal in the HU lanes stand for a fraction of non-replicating dna2-2 and dna2-2 yen1Δ cells?

3. A second key aspect of this manuscript stems from the intrinsic DNA damage sensitivity of dna2-2 cells. Following MMS treatment, dna2-2 cells exhibit some sensitivity in spotting assay (Formosa and Nittis, Genetics, 1999 where dna2-2 is extremely sensitive to a 0.006% MMS treatment). In Fig2.a, dna2-2 allele do not show any sensitivity in spotting assay on plates containing up to 20mM HU (and no sensitivity either at 20mM HU in Fig3.a), whereas it is affected at the same concentration of HU in Fig2.b. This discrepancy requires clarification, as the take home message in Fig2.b is a suppression of dna2-2 HU sensitivity by over-expression of Yen1. Could it be that Fig2.b shows 100mM HU treatment?

4. Replicative stress in dna2-2 expressing cells supposedly increases the occurrence of post-replicative DNA intermediates (inter-sister chromatids link), have the authors analyzed the occurrence of Rad52 foci in dna2-2 and dna2-2 yen1Δ cells? If Rad52 foci can evidence the persistency of inter-sister links, it would be interesting to have some quantitation of these foci in dna2-2 yen1Δ cells expressing various constructs such as Yen1ON (or over-expressing Yen1 and its catalytic version) and monitor Rad52 foci number and pattern.

5. In Fig3.a the loss of Mus81 significantly sensitizes dna2-2 cells to acute replication stress. Nevertheless, the authors claim that Yen1 stand as a major actor in resolution of these intermediates in the context of a helicase-deficient allele of dna2 because Mus81-Mms4 usually take over Yen1 for resolution of these intermediates in a wild-type background. Because the defect of dna2-2 cells triggers G2/M arrest, it would be informative to know whether the cells are still in a window of active Cdk1 (phosphorylating Mms4 for increased resolvase activity) or if cells already committed to FEAR with active Cdc14 and dephosphorylated Mms4. Fig3.b needs to be further substantiated and DAPI staining rather than FACS analyses could more precisely described the cell stage.

Moreover, plating assay could well complement the spotting assay in order to get better resolution in the degree of sensitivity of the various genetic backgrounds used.

6. In this study, a treatment with 200mM of HU precludes dna2-2 cells division (Fig1). The authors then use a dose of 50mM HU (Fig4, 5 and6) for cell-cycle arrest/release and recovery. For clarity, the viability graphs need to be expressed in the same units (Fig1.d , Fig4.b and c ; Fig5.d and Fig6.c). It is not obvious that deletion of rad9 restores dna2-2 cells viability to the same extent as Yen1ON.

7. Furthermore, in Fig5.d, the authors conclude that rad9 deletion restores dna2-2 viability after replicative stress but only marginally when yen1 is absent. In terms of ratio, what is the rescue provided by rad9 deletion? Can the authors analyse the ratio of viability between dna2-2 versus dna2-2 rad9Δ and compare it to those of dna2-2 yen1Δ versus dna2-2 yen1Δ rad9Δ?

8. The authors postulate that Yen1 can compensate dna2-2 cells in non-challenging conditions, but become limiting in presence of increased DNA damage (Fig2.b). According to the Western blot against Yen1ON in Fig6.b, it seems that low level of Yen1ON are sufficient to avoid checkpoint activation (see 120min after recovery). Does over-expressed Yen1 (Fig2.b) still suffer nuclear exclusion by Cdk1 mediated phosphorylation?

Minor concerns:

Is alpha-factor added back to the medium following HU treatment in Fig4.a?

The evidenced genetic interaction between *dna2-2* and *yen1* is temperature sensitive, as synthetic sickness is observed when cells are incubated at 30{degree sign}C while 37{degree sign}C leads to synthetic lethality (Budd et al, PLoS Genetics, 2005). Have the authors tried to perform some DNA damage treatment and rescue experiments using *dna2-2 yen1Δ* cells expressing *Yen1ON* at high temperature?

REVIEWERS' COMMENTS:

Reviewer #1 (Remarks to the Author):

The authors have gone to great lengths to answer previous criticisms, and the manuscript is now significantly more interesting. They recruited an outstanding biochemist to purify the dna2-2 mutant protein and to show that it was ATPase dead but nuclease proficient, as requested. Furthermore, they carried out an interesting 2D gel analysis that showed not only that dna2-2 accumulated incomplete replication intermediates but that they were resolved in the Yen1-On expressing cells. They also carried out many experiments requested by another reviewer that clarified other issues. The manuscript is now acceptable for publication.

Minor points:

1. In the ATPase assay, what does the abscissa represent? What is different at each time point? This is not made clear.
2. In the 2D gel assays, did the aberrant X-intermediates accumulate to higher levels in the dna2-2 yen1 delete double mutant?

Reviewer #2 (Remarks to the Author):

The revised version of the article from Olmezer et al present now compelling evidences for an unanticipated role of Yen1 in the processing of peculiar DNA secondary structures.

Overall, the manuscript has been greatly restructured and provides a smooth and clear investigation of Yen1 requirements in clearing post-replicative damages accumulated in cells expressing a helicase-deficient allele of Dna2 (dna2-2).

The sections unraveling a new function for Yen1 in processing DNA entanglements that differ from Holliday Junction is much clearer and really highlights the crucial role of Yen1 in dealing with a specific type of damages (those arising in cells expressing a Dna2 helicase deficient allele). The fact that Yen1 has a more pronounced effect in dealing with these structures than an activated Mus81-Mms4 stands here as the first report of a prominent function of Yen1 over Mus81-Mms4.

The main concern in the first version of this article stemmed from the lack of evidences about the existence of these replications intermediates (RI) in dna2-2 cells. This is now solved by the 2D gels analyses within the rDNA.

While the authors have replied and substantiate most if not all the points that were raised in their first version of the manuscript, I still would like to raise one minor point about Fig6.

Fig.6 g : have the authors investigated the by 2D gels dna2-2 yen1+ empty vector/Yen1 ON ?

Wouldn't the accumulation of RI be more dramatic in dna2-2 yen1 + empty vector ?

For sure this question should not impinge on further processing this article, I'm still wondering if Fig. 6 g provides the undeniable proof or if I'm missing the point here.

In conclusion, this piece of work reaches now the standards of Nature Communication and greatly deserves to be shared among its readership.

Reviewer #1

Author's response: We thank reviewer #1 for insightful comments and constructive experimental suggestions. Our experimental revisions are detailed point-by-point below.

1. This manuscript claims to show that Dna2 helicase deficiency leads to aberrant replication structures that persist into mitosis if not repaired by Yen1. However, the data at this stage are insufficient to support the claim. First, the authors never show that Dna2 deficiency leads to aberrant intermediates, which they propose are X-shaped structures involving sister chromatid junctions (resembling Yen1 substrates) in mitosis. Fig. 1e, the PFGE experiment, is completely unconvincing, is not quantified, and in any case is not state of the art. The authors need to carry out 2D gels of replication intermediates at a defined genomic locus, a much more precise probe of incomplete DNA replication than HU treatment (see study of Sgs1 by Foiani, Branzei and Rothstein and colleagues, Bernstein et al., 2009). In the absence of Dna2, these gels should show an increase in species that have been shown correspond to various interlinked, unresolved sister chromatid forms and these should increase in the dna2 yen1 mutant (see Gianatassio et al. 2014 NSMB).

Author's response: We have addressed the nature of the aberrant DNA intermediates that accumulate in Dna2 helicase-defective cells, and have characterized the ability of Yen1 to resolve them (see also reviewer #2, point 1) through further experimentation. Following the suggestion of reviewer #1, we removed the PFGE experiment, and instead provide 2D gel analyses of DNA replication intermediates at a defined genomic locus. We have analyzed replication intermediates at the natural replication barrier RFB within the rDNA (**new Fig. 6f**). In good agreement with data from the Campbell lab (Weitao et al., *J. Biol. Chem.* 278, 22513-22522 (2003)), we find that RFB-blocked and converged replication forks persist in G2/M in *dna2-2* cells. As aberrant DNA intermediates that arise in Dna2 helicase-defective cells activate the DNA damage checkpoint (**Figs. 2d, 4a**), affected cells delay progression into mitosis and Yen1 is retained in the cytoplasm (**new Fig. 5a,b**), prohibiting Yen1 from acting on these DNA structures. To see whether Yen1 is able to resolve persistent DNA replication intermediates, we now took advantage of the constitutively active form of Yen1 (Yen1^{on}). Yen1^{on} expression led to a marked reduction of 2D signals for replication intermediates (Y-structures and converged forks) and recombination structures (DNA in the characteristic X-spike), in Dna2 helicase-defective cells (**new Fig. 6g**). Activity towards X-spike DNA was expected, based on the established function of Yen1 in processing recombination intermediates. In contrast, activity towards

Y-DNA and converged forks does not reflect Yen1's canonical role in removing recombination intermediates, identifying persistent, unresolved replication intermediates in Dna2 helicase-defective cells as suitable Yen1 targets. These findings are consistent with Yen1 protecting Dna2 helicase-defective cells from mitotic catastrophe (**Fig. 5d,e,f**) through removing persistent three and/or four-way branched replication intermediates as cells make the transition to anaphase and Yen1 becomes activated. Further support stems from new genetic analyses, in which we show that the synthetic sick relationship between *YEN1* and *dna2-2* is maintained in the absence of homologous recombination. Thus, *dna2-2 rad52Δ* cells are viable, but *dna2-2 rad52Δ yen1Δ* cells are inviable, consistent with a strict requirement for Yen1 to detoxify replication structures, as opposed to recombination structures, in Dna2 helicase-defective cells (**new Fig. 6e**).

2. Second, they claim to show that it is the DNA2 helicase that is responsible for resolving such putative sister chromatid links, and that no role for the DNA2 helicase has been shown before. Ignoring for the moment decades of biochemistry that show that there is crosstalk between the helicase and nuclease, probably through shared substrate binding sites, they use only a single mutant, dna2-2. Contrary to the author's claim, it is not known if dna2-2 protein is deficient in nuclease. Although dna2-2 maps in the helicase domain, it maps in a motif involved in other helicases in DNA binding, which may affect both helicase and nuclease. The dna2-2 protein has never been purified, so it has never been shown to have intact nuclease. The authors cite Zhu et al to support their claim, but that reference is insufficient since Zhu et al. state that dna2-2 was shown to have intact nuclease by three references. However, none of these papers reports on purification and assay of dna2-2.

Furthermore, Zhu et al does not show that dna2-2 protein at normal levels can support resection, since they only show that overproducing dna2-2 restores resection in dna2 pif1 delete. Even overexpression of dna2-1, a temperature sensitive mutant and which is deficient in both helicase and nuclease, suppresses dna2-1 phenotypes. You simply need more enzyme in these hypomorphs. The authors must express and purify dna2-2 and show that nuclease is intact if they are to keep this claim that the helicase is behind the defects observed. Alternatively, they should use a mutant defective only in helicase, such as K1080A. Alternatively drop the claim about the helicase and just say Dna2.

Author's response: We agree to the uncertainties regarding the effect of the R1253Q mutation (*dna2-2*) on the enzymatic activities of Dna2, and have set out to conduct the first biochemical analysis of this mutant (**new Fig. 1a-f, Supplementary Fig. 1**). We have

purified Dna2 R1253Q following overexpression in yeast and subjected the protein to a variety of ATPase, helicase, and nuclease assays, side-by-side with the wild-type protein and previously characterized Dna2 nuclease and helicase mutants. The results (described in detail in the first subsection of RESULTS, page 5, line 121 – page 7, line 154) now clearly show that Dna2 R1253Q is as effective as wild-type as a nuclease, but completely devoid of ATPase/helicase activity. These are important findings and we thank reviewer #1 for pointing this out. We now unambiguously relate *dna2-2* phenotypes and the functional interplay between Dna2 and Yen1 to Dna2's helicase activity.

3. Third, the YEN1-On experiment is interesting and an interesting interpretation is given. However, it is not the only interpretation. The YEN1 might suppress defects in Okazaki fragment processing when it is expressed in S phase, just as FEN1 and EXO1 overexpression suppress dna2 mutations. The authors should test if YEN1-On also suppresses a rad27 mutant to see if YEN1 can also function in OFP. Even if they do not find suppression, they cannot rule out a role in OFP. This is why it is crucial to actually demonstrate that junctional replication intermediates accumulate and are the culprit in dna2 and in dna2 yen1 double mutants (point 1).

Author's response: We share the concern that the proposed role of Dna2 in Okazaki fragment processing must be taken into account. We have now assessed the potential of Yen1 and Yen1^{on} to alleviate *rad27* phenotypes genetically. In contrast to *dna2-2* defects, Yen1/Yen1^{on} proved ineffective in ameliorating *rad27* defects (**new Supplementary Fig. 7a-c**). This suggests that the suppression of DNA damage checkpoint activation and increased cell survival brought about by Yen1^{on} expression in G2/M in *dna2-2* cells following acute replication stress is not related to Yen1 acting to remove unprocessed Okazaki fragments (**Supplementary Fig. 6**). At the same time, we detect replication intermediates as viable targets for Yen1 in Dna2 helicase-defective cells by 2D gel electrophoresis (please see point 1). Finally, our biochemical analysis of purified Dna2 R1253Q shows that the mutant protein is proficient in cleaving 5'-flaps covered with RPA (**new Fig. 1f**), which suggests that the proposed function of Dna2 in Okazaki fragment processing is intact in *dna2-2* cells. These new results are discussed in the DISCUSSION of the revised manuscript (page 18, line 457 – page 19, line 468):

“The ability of Yen1 to cut 5'-flaps is shared with all Rad2/XPF superfamily nucleases, including Rad27. This raises the formal possibility that Yen1 might support Rad27 in Okazaki fragment processing, and that its role in safeguarding Dna2 helicase-defective

cells might pertain to the proposed involvement of Dna2 in this process. In line with previous findings⁵⁸, we find no genetic indication that Yen1 might support Rad27 functions *in vivo*. Thus, deletion of *YEN1* did not aggravate the temperature-sensitive growth defect and MMS sensitivity associated with loss of Rad27, nor did overexpression of Yen1 or Yen1^{on} alleviate these phenotypes (Supplementary Fig. 7). In light of this, and our biochemical analyses showing that Dna2 R1253Q is fully proficient in removing RPA-covered 5'-flaps (Fig. 1f) that might become refractory to cleavage by Rad27 *in vivo*, we do not anticipate Okazaki fragment processing problems in *dna2-2* cells, or, should they exist, that Yen1 would be in a position to ameliorate such problems.”

4. Suppression by rad9 delete is complicated. It has also been shown twice before, and these papers, one of which may not have been published when this one was submitted, are not cited. Rad9 is an inhibitor of resection and must be removed by Fun30, which complicates their model.

Author’s response: We agree that the function of Rad9 in the regulation of end-resection represents a complication in the interpretation of phenotypes that result from a deletion of *RAD9*. Yet, we propose that the suppression of *dna2-2* defects relates first and foremost to the checkpoint function of Rad9 (**Fig. 7b**). In the revised manuscript, we now show that Dna2 R1253Q is as effective as wild-type in degrading tailed DNA substrates (**Fig. 1d,e**). This, together with a report showing that the *dna2-2* allele can support end-resection *in vivo* (Zhu *et al.*, *Cell* 134, 981-994, (2008)), suggests that *dna2-2* phenotypes are not related to an end-resection problem that might be alleviated by loss of Rad9. Consistently, *YEN1* shows no synthetic sickness with deletion of *SGS1* (Blanco *et al.*, *DNA Repair* 9, 394-402 (2010)), the helicase that cooperates with the Dna2 nuclease in DNA end-resection. This is now specifically outlined in the DISCUSSION (page 19, lines 469-479):

“It is not immediately obvious how a Yen1 substrate might arise from a potential defect of *dna2-2* cells in DNA double-strand break repair. However, replication stress can lead to DNA breaks, so it might be conceivable that defects in *dna2-2 yen1Δ* cells reflect a functional overlap between Yen1 and the role of Dna2 in DNA end-resection. This seems unlikely, however, because of the functional redundancy that exists between the Exo1 and Dna2-dependent end-resection pathways⁵⁹. Furthermore, we find that Dna2 R1253Q is proficient in degrading tailed DNA substrates (Fig. 1d,e), consistent with observations that plasmids containing *dna2-2*, but not a *dna2* nuclease-defective allele, could

complement the DNA end-resection phenotype of Dna2-deficient cells *in vivo*⁵⁹. Finally, *YEN1* exhibits no synthetic sick interaction with *SGS1*⁴⁴, the partner helicase of the Dna2 nuclease in DNA end-resection.”

Additional support for the importance of Rad9’s role in checkpoint signaling and cell cycle arrest, as opposed to end-resection control, for loss of viability in *dna2-2* cells relates to the following observations: A complete restoration of viability after acute replication stress can be achieved by the deletion of *RAD9* (**Fig. 5f**). However, this effect is strictly dependent upon the activity of Yen1 (**Fig. 5f**), whose activation is coupled to anaphase onset, an event that is blocked by Rad9-dependent DNA damage signaling in *dna2-2* cells (**Figs. 2d,e, 4a, and 5c**). Consistently, we now demonstrate that Yen1 is retained in the cytoplasm throughout HU-induced G2/M arrest in *dna2-2* cells, unable to access nuclear DNA lesions (**new Fig. 5a,b**). When checkpoint-deficient *dna2-2 rad9Δ* cells enter anaphase, Yen1 is required to clear the two segregating masses of nuclear DNA from connecting DNA bridges that preclude mitosis in *dna2-2 rad9 yen1Δ* cells. This shows that physical sister-chromatid links constitute a critical lesion in Dna2 helicase-defective cells (**Fig. 5e**). While their resolution is linked to Yen1 activation in anaphase, artificial activation of Yen1 in G2/M allows Yen1 to act on DNA lesions in checkpoint-proficient Dna2 helicase-defective cells prior to anaphase, suppressing DNA damage checkpoint activation, and improving survival (**Supplementary Fig. 6**). Expression of constitutively active Yen1^{on} further leads to a reduction in replication intermediates that accumulate in *dna2-2* cells (**new Fig. 6g**). Together, these experiments demonstrate that a key event in rescuing Dna2 helicase-defective cells from terminal G2/M arrest and mitotic problems arising from post-replicative chromosomal DNA links, is overcoming the inhibitory effect that Rad9-dependent checkpoint signaling has on Yen1 activation.

Failure to activate Yen1 for the resolution of toxic, but not inherently lethal, DNA intermediates in Dna2 helicase-defective cells provides a molecular explanation for the conspicuous suppression of *dna2* phenotypes by disruption of *RAD9*, reports of which go back almost 20 years. Fiorentino & Crabtree (*Mol. Biol. Cell* 8, 2519-2537 (1997)) found that deletion of *RAD9* improved the viability of cells with an unspecified, temperature sensitive *DNA2* mutation from 2.77% to 6.53% after a 4 h-exposure to the restrictive temperature of 37 °C (compared to a viability of 94.8% for the *rad9* control strain). Similarly, Formosa & Nittis (*Genetics* 151, 1459-1470 (1999)) reported residual survival (approx. 3% viability) of otherwise temperature-lethal *dna2* alleles when they were combined with a deletion of *RAD9*. More recently, Judith Campbell and colleagues (Budd

et al., *Cell cycle* 10, 1690-1698 (2011)) reported that the lethality of a *DNA2* deletion is suppressed by mutation or loss of Rad9. The defects in these strains, in particular those of the *dna2Δ* strain, are likely to be more complex and severe than in case of the *dna2-2* strain with a specific inactivation only of the helicase activity, and not the nuclease activity; this probably explains the limited improvements in viability upon deletion of *RAD9* in some previous studies. Nonetheless, the failure to activate Yen1 after Rad9-dependent checkpoint activation might well contribute to the detriment of *dna2Δ* and temperature-lethal variants of Dna2 at the restrictive temperature, and thus explain, at least in part, suppression of growth defects by deletion of *RAD9*. We thank the reviewer for flagging this up and now make reference to the aforementioned papers in the revised version of the manuscript. Please see RESULTS, page 13, lines 316-331:

“When we determined the effect of checkpoint disruption on cell viability, we found that deletion of *RAD9* increased the viability of Dna2 helicase-defective cells after acute replication stress treatment 3-fold, reaching levels very similar to those observed for the *rad9Δ* control strain. In the absence of *YEN1*, viability was also improved, but did not reach more than ~14% of the viability of the *rad9Δ* control (Fig. 5f). Checkpoint activation in Dna2 helicase-defective cells after acute replication stress therefore appears futile, and eliminating the G2/M checkpoint enabled a highly effective Yen1-dependent survival pathway, while allowing a small subset of cells to survive in a Yen1-independent manner. As expected, checkpoint disruption had no beneficial effect when cells were exposed to chronic replication stress (Supplementary Fig. 5), consistent with improved survival being linked specifically to allowing Yen1 access to post-replicative lesions in Dna2 helicase-defective cells after acute replication stress. These observations resonate with previous findings showing that the lethality of some temperature-sensitive *dna2* alleles, and of *dna2Δ*, can be suppressed by deleting *RAD9*^{15,49,50}, linking this phenomenon, at least for the Dna2 helicase-defective *dna2-2* allele, to the removal of the inhibitory effect of the G2/M DNA damage checkpoint on Yen1 activation.”

Reviewer #2:

In their manuscript entitled "Dna2 nuclease helicase and Holliday junction resolvase Yen1 provide a two-tiered response to replication stress", Ölmezer and colleagues investigate the interplay between Dna2 and Yen1 and uncover an unexpected contribution of the

Yen1 resolvase in dealing with a specific type of DNA damage in order to ensure genome stability.

Dna2 is an essential DNA nuclease helicase involved in pleiotropic cell vital processes ranging from Okazaki fragments maturation to DNA repair. The gene coding for Yen1, a non-essential structure specific nuclease targeting Holliday junctions in late G2/M phase of the cell cycle, was identified as a synthetic lethal interacting gene of the dna2-helicase deficient (dna2-2) allele.

In this study, cells constitutively expressing the helicase deficient allele dna2-2, present an increased G2/M-arrested subpopulation. Deletion of yen1 increases this G2/M subpopulation (as well as the doubling time), but more importantly, decreases the viability of dna2-2 cells. The authors raise here the idea that dna2-2 cells require Yen1 to resolve persistent DNA links in dna2-2 cells and undergo mitosis.

dna2-2 and dna2-2 yen1Δ cells undergoing recovery after replication stress (50mM HU allowing some dna2-2 cells to undergo mitosis after recovery) display similar replication profiles as well as identical Rad53 activation kinetics, further reinforcing the model in which dna2-2 cells suffer late replication problems rather than S-phase defects. Furthermore, this assay clearly shows a checkpoint reactivation (measured by Rad53 phosphorylation) in dna2-2 and dna2-2 yen1Δ at the time mitosis. Strikingly, deletion of rad9 suppresses dna2-2 checkpoint reactivation and restores the viability of dna2-2 after replicative stress in a yen1 dependent fashion. Here the authors demonstrate that dna2-2 G2/M arrest stands more as an impediment for Yen1 activity. This hypothesis got further weight in the recovery from replicative stressed followed by nocodazole arrest, where the authors unveiled that Yen1 activation per se is required to resolve DNA intermediates accumulating in dna2-2 cell.

This sheds new lights in the function of Yen1 in a peculiar replicative stress condition (mediated by the dna2 helicase-deficient allele, dna2-2) and will certainly be of interest for the readership of Nature Communications. Nevertheless, in its actual shape, the manuscript suffers of some overstatements and would be probably gain in strength with further experimental evidences as well as with some clarifications.

Major concerns:

- 1. The key point of this work is the ability of Yen1 to resolve secondary DNA structures arising in cells expressing dna2 helicase deficient allele (dna2-2) following or not replication stress (HU treatment). Despite being highly reproducible, the PFGE quantitation (Fig1.e) is only an indirect evidence for the presence of post-replicative DNA*

inter-sister chromatids, and probably 2D gels analyses would be more informative. Quantitation of these intermediates in replicative stress conditions, in presence or absence of Yen1 could certainly strengthen the message.

Author's response: We thank reviewer #2 for highlighting the importance of our work. We have also taken the criticism to heart, and have now addressed the points raised through further experiments.

We agree that the use of 2D gel analyses of DNA derived from a defined genomic locus provides a more informative assessment of branched DNA intermediates that accumulate in *dna2-2* cells. We have therefore replaced the PFGE experiment (see also reviewer #1, point 1) with new 2D gel analyses of the rDNA. The natural replication fork barrier (RFB) within the rDNA provides a site of anticipated replication stalling, allowing us to study the effects of the Dna2 helicase and Yen1 at a specific genomic locus. As presented in **new Fig. 6f,g**, and consistent with a previous report (Weitao *et al.*, *J. Biol. Chem.* 278, 22513-22522 (2003)), *dna2-2* cells exhibit increased levels of replication intermediates (converged and RFB-stalled forks, Y-structures), as well as recombination intermediates (X-DNA), which provides direct evidence for an accumulation of unresolved DNA links between sister chromatids in Dna2 helicase-defective cells. Importantly, in the presence of activated Yen1 (expression of constitutively active variant Yen1^{on}, circumventing the caveat that aberrant DNA intermediates in *dna2-2* activate the G2/M checkpoint and therefore preclude the nuclear accumulation and activation of Yen1; please see also point 8), these intermediates, with the exception of RFB-stalled forks, were markedly reduced. These results show that, while not immediately targeting RFB-stalled forks, Yen1 removes converged forks, fork Y-structures, and (as expected) X-DNA.

While the dead-end replication structures as well as the recombination intermediates detected by 2D gel analyses constitute inter-sister chromatid links that might preclude chromosome segregation in Dna2 helicase-defective cells, we provide further evidence that persistent replication structures critically require detoxification by Yen1. Thus, we find that Mus81-Mms4, the major nuclease for removing recombination intermediates, is present in its activated form throughout replication stress-induced G2/M arrest in Dna2 helicase-defective cells (**new Supplementary Fig. 8**). Yet, the presence of Mus81-Mms4 does not circumvent the requirement for Yen1 in Dna2 helicase-defective cells. Moreover, new genetic analyses demonstrate that *dna2-2 rad52Δ* cells are viable, but *dna2-2 rad52Δ yen1Δ* cells are not (**new Fig. 6e**), showing

that Dna2 helicase-defective cells depend upon Yen1 for viability even in the absence of homologous recombination. This is consistent with the ability of Yen1 to target replication intermediates, as detected by our 2D gel analyses, being critical for protecting Dna2 helicase-defective cells from sister chromatid non-disjunction and mitotic catastrophe (**Fig. 7b**).

2. Does the faint DNA signal in the HU lanes stand for a fraction of non-replicating dna2-2 and dna2-2 yen1Δ cells?

Author's response: Yes, this observation is correct. A clarification in the revised version has become obsolete because the PFGE experiment in question has been replaced by 2D gel analyses (please see point 1).

3. A second key aspect of this manuscript stems from the intrinsic DNA damage sensitivity of dna2-2 cells. Following MMS treatment, dna2-2 cells exhibit some sensitivity in spotting assay (Formosa and Nittis, Genetics, 1999 where dna2-2 is extremely sensitive to a 0.006% MMS treatment). In Fig2.a, dna2-2 allele do not show any sensitivity in spotting assay on plates containing up to 20mM HU (and no sensitivity either at 20mM HU in Fig3.a), whereas it is affected at the same concentration of HU in Fig2.b. This discrepancy requires clarification, as the take home message in Fig2.b is a suppression of dna2-2 HU sensitivity by over-expression of Yen1. Could it be that Fig2.b shows 100mM HU treatment?

Author's response: The higher apparent sensitivity to HU in **former Fig. 2b** is due to the use of selective synthetic medium (SC ura-; for selection for the Yen1 complementation plasmid) as opposed to rich medium. We have now repeated this complementation assay using YPAD medium (**new Fig. 3b**). This harmonizes the apparent HU sensitivity levels exhibited by *dna2-2* mutant cells in this assay with the other drop assays presented throughout the paper without changing the results: plasmid-based expression of Yen1 significantly suppresses the HU-sensitive growth of *dna2-2 yen1Δ* cells and also, to some extent, of *dna2-2* cells, and these effects strictly depend on the nuclease activity of Yen1 (**new Fig. 3b**).

We routinely observe slower growth and higher apparent sensitivity to HU when growing *dna2-2* mutants, or other DNA replication/repair mutants, on SC dropout medium rather than YPAD. The use of CEN plasmids, however, provides us with the opportunity to carry out all analyses on YPAD, since the rate of CEN plasmid loss per generation is very low, even in the absence of selection. Consistent use of YPAD now

helps avoid any confusion that might arise due to differences in growth of *dna2-2* mutant cells on synthetic versus rich medium, and we thank reviewer #2 for pointing this out.

4. Replicative stress in dna2-2 expressing cells supposedly increases the occurrence of post-replicative DNA intermediates (inter-sister chromatids link), have the authors analyzed the occurrence of Rad52 foci in dna2-2 and dna2-2 yen1Δ cells? If Rad52 foci can evidence the persistency of inter-sister links, it would be interesting to have some quantitation of these foci in dna2-2 yen1Δ cells expressing various constructs such as Yen1ON (or over-expressing Yen1 and its catalytic version) and monitor Rad52 foci number and pattern.

Author's response: We have now expressed a functional version of homologous recombination mediator Rad52 tagged with yellow fluorescent protein (Feng *et al.*, *DNA Repair* 6, 27–37 (2007)) to assess Rad52 foci in Dna2 helicase-defective cells. We find that Rad52 foci are markedly increased, in particular in G2/M (**new Fig. 6b**). We have not found evidence for a significant additional change in the number of Rad52 foci due to the absence of Yen1, or upon plasmid-based expression of Yen1/nuclease-dead Yen1 in *dna2-2* cells (data not shown). Elevated S and G2/M phase Rad52 foci (**new Fig. 6b**) are consistent with a higher incidence of homologous recombination repair in Dna2 helicase-defective cells, in line with the synthetic sick phenotype/synthetic hypersensitivity to replication stress that we find in *dna2-2 rad52Δ* cells (**Fig. 6a**). Downstream, these homologous recombination events may lead to the formation of joint molecules, contributing to sister chromatid entanglements in *dna2-2* cells. The majority of these intermediates, should they require nucleolytic processing, will be subject to resolution by Mus81-Mms4 in G2/M; consistently, deletion of *MUS81* caused additive replication stress sensitivity with *dna2-2*. Importantly, however, *dna2-2 mus81Δ* cells do not recapitulate the severe synthetic sickness caused by loss of *YEN1* in the *dna2-2* background (**Fig. 6c**, **new Fig. 6d**). Moreover, the synthetic sick relationship between *dna2-2* and *YEN1* is maintained in the absence of homologous recombination (**new Fig. 6e**). Therefore, those toxic DNA intermediates that strictly require Yen1 for detoxification in *dna2-2* cells are not recombination intermediates, and Rad52 foci are not a direct measure for them. A parallel requirement for Rad52-dependent homologous recombination and the Dna2 helicase activity for cells to respond to replicative stress is an important point, and we thank the reviewer for the suggestion to develop this point further (please see also point 5).

5. In Fig3.a the loss of Mus81 significantly sensitizes *dna2-2* cells to acute replication stress. Nevertheless, the authors claim that Yen1 stand as a major actor in resolution of these intermediates in the context of a helicase-deficient allele of *dna2* because Mus81-Mms4 usually take over Yen1 for resolution of these intermediates in a wild-type background. Because the defect of *dna2-2* cells triggers G2/M arrest, it would be informative to know whether the cells are still in a window of active Cdk1 (phosphorylating Mms4 for increased resolvase activity) or if cells already committed to FEAR with active Cdc14 and dephosphorylated Mms4. Fig3.b needs to be further substantiated and DAPI staining rather than FACS analyses could more precisely described the cell stage.

Author's response: We have now analyzed the phosphorylation status of Mms4 in *dna2-2* and wild-type cells exposed to acute HU treatment, leading to G2/M arrest in *dna2-2* cells. In wild-type cells, Mms4 is present in its active, hyperphosphorylated form from late S to G2/M, before phosphorylation diminishes in M phase and the subsequent G1 phase. In contrast, Mms4 is hyperphosphorylated in *dna2-2* cells with kinetics similar to wild-type, but then phosphorylation is maintained as the mutant cells arrest at G2/M (**new Supplementary Fig. 8**). This is consistent with the presence of a DNA substrate that is refractory to activated Mus81-Mms4 and requires Yen1 for resolution. We now further demonstrate that the synthetic sick relationship between *dna2-2* and *YEN1* is maintained in the absence of homologous recombination (**new Fig. 6e**), showing that DNA structures that strictly required resolution by Yen1 in *dna2-2* cells arise independently of homologous recombination. Hence, Yen1 acts along a pathway distinct from canonical Holliday junction resolution, and in a manner that is non-redundant with Mus81-Mms4. This leads to an unprecedented scenario, where loss of Yen1 is more detrimental than loss of Mus81-Mms4, with *dna2-2 yen1Δ* double mutant cells exhibiting poorer survival than *dna2-2 mus81Δ* cells (**Fig. 6c, new Fig. 6d**). Nonetheless, a significant sensitization of *dna2-2* cells by deletion of *MUS81* is expected since *dna2-2* cells exhibit increased levels of homologous recombination (**new Fig. 6b**), and Mus81-Mms4 is the major nuclease removing homologous recombination intermediates between sister chromatids.

As suggested, we have also added an analysis of cells after acute HU treatment stained with DAPI, and in addition have analyzed cells for the localization of a green fluorescent protein-tagged version of Yen1 (Yen1-EGFP). We find that cells arrest with a single mass of nuclear DNA near the bud-neck and Yen1 homogeneously located in the cytoplasm, consistent with a pre-anaphase arrest (**new Fig. 5a,b**).

Taken together, these new results demonstrate that, following replication stress, *dna2-2* cells arrest prior to commitment to FEAR and Cdc14 activation, with hyperphosphorylated-active Mus81-Mms4 and cytoplasmic-inactive Yen1. These points have been incorporated in our model presented in **Fig. 7**; **former Fig. 3b** has been omitted due to redundancy.

Moreover, plating assay could well complement the spotting assay in order to get better resolution in the degree of sensitivity of the various genetic backgrounds used.

Author's response: Plating assays have been added as requested (presented in **new Fig. 6d**).

6. In this study, a treatment with 200mM of HU precludes dna2-2 cells division (Fig1). The authors then use a dose of 50mM HU (Fig4, 5 and6) for cell-cycle arrest/release and recovery. For clarity, the viability graphs need to be expressed in the same units (Fig1.d , Fig4.b and c ; Fig5.d and Fig6.c). It is not obvious that deletion of rad9 restores dna2-2 cells viability to the same extent as Yen1ON.

Author's response: All cell viability graphs (**revised Figs. 2a, 2f, 4c, 5f**) now uniformly express strain viability relative to the relevant control strains (wild-type or *rad9Δ*). Regarding the experiment with transient Yen1^{on} expression in *dna2-2* cells, we cannot draw meaningful parallels to the experiments with *RAD9*-deleted strains. Yen1^{on} expression causes well-documented defects related to untimely activation of the enzyme (Blanco *et al.*, *Mol. Cell* 54, 94–106 (2014); Chan & West, *Nat. Commun.* 5, 4844 (2014); Eissler *et al.*, *Mol. Cell* 54, 80–93 (2014); Matos *et al.*, *Cell Rep.* 4, 76–86 (2013); Szakal & Branzei, *EMBO J.* 32, 1155–1167 (2013)), so that in this experiment, viability levels are the result of mixed beneficial and detrimental effects. In this case, the relevant result is therefore the marked relative increase in cell viability (approx. threefold) that we document in comparison with a control strain with empty vector (see revised manuscript, page 14, lines 336-343).

7. Furthermore, in Fig5.d, the authors conclude that rad9 deletion restores dna2-2 viability after replicative stress but only marginally when yen1 is absent. In terms of ratio, what is the rescue provided by rad9 deletion? Can the authors analyse the ratio of viability between dna2-2 versus dna2-2 rad9Δ and compare it to those of dna2-2 yen1Δ versus dna2-2 yen1Δ rad9Δ?

Author's response: The description of the beneficial effect of loss of *RAD9* for the viability of *dna2-2 yen1Δ* cells as *marginal* in the former version of the manuscript was inappropriate. We thank the reviewer for spotting this inaccuracy. In fact, given that the viability of *dna2-2 yen1Δ* is virtually zero after replication stress, the fold-change in viability upon deletion of *RAD9* is rather substantial. However, the triple mutant reaches a viability of only about 14% of the *rad9Δ* control. In contrast, deletion of *RAD9* restores full viability to *dna2-2 cells*, i.e. a viability level similar to the *rad9Δ* control. This is the key point we intend to highlight, as it indicates the existence of a non-fatal lesion that is effectively resolved across the cell population when Yen1 is enabled to act. In *dna2-2 yen1Δ* cells, however, the vast majority of cells cannot detoxify these Yen1-targeted DNA lesions, so that – despite a gain in survival – viability remains comparatively low. We have now rephrased the respective section accordingly (RESULTS, page 13, lines 316-323):

“When we determined the effect of checkpoint disruption on cell viability, we found that deletion of *RAD9* increased the viability of Dna2 helicase-defective cells after acute replication stress treatment 3-fold, reaching levels very similar to those observed for the *rad9Δ* control strain. In the absence of *YEN1*, viability was also improved, but did not reach more than ~14% of the viability of the *rad9Δ* control (Fig. 5f). Checkpoint activation in Dna2 helicase-defective cells after acute replication stress therefore appears futile, and eliminating the G2/M checkpoint enabled a highly effective Yen1-dependent survival pathway, while allowing a small subset of cells to survive in a Yen1-independent manner.”

8. *The authors postulate that Yen1 can compensate dna2-2 cells in non-challenging conditions, but become limiting in presence of increased DNA damage (Fig2.b). According to the Western blot against Yen1ON in Fig6.b, it seems that low level of Yen1ON are sufficient to avoid checkpoint activation (see 120min after recovery). Does over-expressed Yen1 (Fig2.b) still suffer nuclear exclusion by Cdk1 mediated phosphorylation?*

Author's response: We have now overexpressed a functional (**new Supplementary Fig. 4a**) version of Yen1, tagged with enhanced green fluorescent protein (EGFP), in *dna2-2* cells. The fusion protein exhibited the characteristic (Blanco *et al.*, *Mol. Cell* 54, 94–106 (2014); García-Luis *et al.*, *Cell Cycle* 13, 1392–1399 (2014); Eissler *et al.*, *Mol. Cell* 54, 80–93 (2014); Kosugi *et al.*, *Proc. Natl. Acad. Sci.* 106, 10171–10176 (2009)) cell cycle-dependent subcellular localization pattern (**new Fig. 5a,b**). After acute HU treatment, cells accumulated as large-budded G2/M cells – then, *dna2-2* cells homogenously

exhibited cytoplasmic Yen1-EGFP signals, indicating that even after overexpression, Yen1 remains excluded from the nucleus. Yen1-EGFP was only ever seen accumulated in the nucleus after anaphase entry, i.e. in mitotic, double-nucleated cells. While these experiments cannot rule out that increased levels of Yen1 upon overexpression might (eventually) facilitate some extent of nuclear entry, they show that *dna2-2* cells arrest before anaphase, as expected due to unscheduled DNA damage checkpoint activation in G2/M (**Fig. 4a**), and that Yen1 – at least initially (we examined samples up to 4 h post-treatment) – is largely excluded from the nucleus.

Minor concerns

Is alpha-factor added back to the medium following HU treatment in Fig4.a?

Author's response: Yes, this is indicated in the experimental scheme in **Fig. 4a**; for clarity this is now additionally stated in legend to **Fig. 4** in the revised manuscript.

The evidenced genetic interaction between dna2-2 and yen1 is temperature sensitive, as synthetic sickness is observed when cells are incubated at 30{degree sign}C while 37{degree sign}C leads to synthetic lethality (Budd et al, PLoS Genetics, 2005). Have the authors tried to perform some DNA damage treatment and rescue experiments using dna2-2 yen1Δ cells expressing Yen1ON at high temperature?

Author's response: In our hands, elevated temperature impairs the growth of *dna2-2* and *dna2-2 yen1Δ* to some extent, but we find no evidence for temperature-lethality (**new Supplementary Fig. 2c**; see also **new Supplementary Fig. 7b**). A statement is now included in the RESULTS to make this clear (page 7, lines 171-174):

“Contrary to a reported temperature-dependent lethal interaction between *YEN1* and *DNA2*³⁵, we found double mutant cells were viable at elevated temperature (37°C) (Supplementary Fig. 2c), although doubling times for *dna2-2* and *dna2-2 yen1Δ* were further increased by ~20 min and ~5 min, respectively.”

Reviewer #1 (Remarks to the Author):

The authors have gone to great lengths to answer previous criticisms, and the manuscript is now significantly more interesting. They recruited an outstanding biochemist to purify the dna2-2 mutant protein and to show that it was ATPase dead but nuclease proficient, as requested. Furthermore, they carried out an interesting 2D gel analysis that showed not only that dna2-2 accumulated incomplete replication intermediates but that they were resolved in the Yen1-On expressing cells. They also carried out many experiments requested by another reviewer that clarified other issues. The manuscript is now acceptable for publication.

Author's response: We thank reviewer #1 for highlighting the importance of the biochemical work, 2D gel analyses, and further experimental evidence that has been added the manuscript. Minor points raised are addressed in detail below.

Minor points:

1. In the ATPase assay, what does the abscissa represent? What is different at each time point? This is not made clear.

Author's response: The kinetic plot in Fig. 1c is derived from an ATPase assay that is based on the conversion of phosphoenol pyruvate to pyruvate when ADP, generated by Dna2, is regenerated to ATP by pyruvate kinase. The pyruvate is subsequently converted to lactate upon oxidation of one equivalent of NADH to NAD⁺ by lactate dehydrogenase. This oxidation of NADH is followed spectrophotometrically over time and the rate of change of absorbance at 340 nm is proportional to rate of ATP hydrolysis in a given time interval (Kreuzer KN & Jongeneel CV, *Methods Enzymol.* 100, 144-160 (1983); Kowalczykowski, SC & Krupp RA. *J. Mol. Biol.* 193, 97-113 (1987); Levikova *et al.*, *Proc. Natl. Acad. Sci.* 110, E1992-2001 (2013)), as shown in Fig. 1c. We apologize for not making this sufficiently clear and have now added additional information on the ATPase assay to the Materials and Methods section in the revised manuscript (line 705):

“In brief, we used an assay that couples ATP hydrolysis to oxidation of NADH to determine the rate of ATP hydrolysis by Dna2 variants by following the decrease in optical absorbance by NADH at 340 nm over time. Kinetic plots of ATP hydrolysis were derived by calculating the amount of ATP hydrolyzed per time interval.”

2. In the 2D gel assays, did the aberrant X-intermediates accumulate to higher levels in the dna2-2 yen1 delete double mutant?

Author's response: Since *dna2-2 yen1*Δ do not synchronize as well as *dna2-2* cells, which complicates cell-cycle staged 2D gel experiments, we have not analyzed the double mutant cells at the same level of detail as single mutant cells. However, in preliminary experiments, we have never seen evidence for an accumulation of replication or recombination intermediates in double mutant *dna2-2 yen1*Δ cells above the levels seen in the *dna2-2* single mutant. This is perhaps not surprising as Yen1 is activated and nuclear only after anaphase entry (Fig. 5a,b) (West et al., *Cold Spring Harb. Symp. Quant. Biol.* 80, 103-109 (2015)), and as a result it has no bearing on the levels of *de novo dna2-2*-related lesions in S and G2 phase. Thus, any observable accumulation of aberrant DNA intermediates arising from Dna2 helicase deficiency during replication relates to *dna2-2*, and is unaffected by endogenous Yen1.

Reviewer #2 (Remarks to the Author):

The revised version of the article from Olmezer et al present now compelling evidences for an unanticipated role of Yen1 in the processing of peculiar DNA secondary structures.

Overall, the manuscript has been greatly restructured and provides a smooth and clear investigation of Yen1 requirements in clearing post-replicative damages accumulated in cells expressing a helicase-deficient allele of Dna2 (dna2-2).

The sections unraveling a new function for Yen1 in processing DNA entanglements that differ from Holliday Junction is much clearer and really highlights the crucial role of Yen1 in dealing with a specific type of damages (those arising in cells expressing a Dna2 helicase deficient allele). The fact that Yen1 has a more pronounced effect in dealing with these structures than an activated Mus81-Mms4 stands here as the first report of a prominent function of Yen1 over Mus81-Mms4.

The main concern in the first version of this article stemmed from the lack of evidences about the existence of these replications intermediates (RI) in dna2-2 cells. This is now solved by the 2D gels analyses within the rDNA.

While the authors have replied and substantiate most if not all the points that were raised in their first version of the manuscript, I still would like to raise one minor point about Fig6.

Fig.6 g : have the authors investigated the by 2D gels dna2-2 yen1+ empty vector/Yen1 ON ? Wouldn't the accumulation of RI be more dramatic in dna2-2 yen1 + empty vector ?

For sure this question should not impinge on further processing this article, I'm still wondering if Fig. 6 g provides the undeniable proof or if I'm missing the point here.

In conclusion, this piece of work reaches now the standards of Nature Communication and greatly deserves to be shared among its readership.

Author's response: We are grateful for the positive assessment of our revised manuscript for publication by reviewer #2.

Fig.6 g : have the authors investigated the by 2D gels dna2-2 yen1+ empty vector/Yen1 ON ?

Author's response: For reasons outlined above (please see reviewer #1, point 2), we have not analyzed *dna2-2 yen1Δ* double mutant cells by 2D gel analysis to any great depth at present. For the Yen1^{on} experiment, designed to show in principle that Yen1 can target recombination and replication structures that persist in Dna2 helicase-defective cells, we deliberately chose to use *dna2-2* single mutant cells. In our opinion, *dna2-2* single mutant

cells are best suited because they do not carry an additional genetic defect (*yen1Δ*) that might give rise to DNA structures detectable by 2D gel analysis. With endogenous Yen1 present, chromosome entanglements in *dna2-2* cells are resolved at mitosis, avoiding cell death and potential secondary DNA damage. At the same time, endogenous Yen1 has no effect on replication-associated structures arising due to Dna2 helicase deficiency because of the temporal separation of the activities of Dna2 (S-G2 phase of the cell cycle) and Yen1 (M phase). Consistently, we detect elevated levels of persistent replication and recombination intermediates in *dna2-2* cells in the presence of endogenous Yen1, which we find are reduced upon expression of constitutively active Yen1^{on} (Fig. 6f,g).

To make this distinction between endogenous Yen1 and Yen1^{on} in our experimental set-up clearer, we amended the manuscript text on page 16. The sentence starting in line 496 now reads:

“To see if Yen1 targets aberrant replication intermediates that persist in Dna2 helicase-defective cells, we expressed, in asynchronous *dna2-2* cultures, constitutively active Yen1^{on} (Fig. 6g), allowing us to monitor Yen1 actions prior to the activation of endogenous Yen1 and chromosome segregation in M phase.”; and the sentence starting in line 510: “These results suggest that endogenous Yen1, activated upon anaphase entry, uniquely resolves persistent replication fork/converging fork structures to disentangle underreplicated nascent sister chromatids when Dna2 helicase-defective cells enter mitosis, thereby safeguarding chromosome segregation and enabling viable mitotic exit.”

Wouldn't the accumulation of RI be more dramatic in dna2-2 yen1 + empty vector ?

Author's response: Lesions derived from Dna2 helicase deficiency arise at the same level in *dna2-2* or *dna2-2 yen1Δ* cells in a given cell cycle. Whether and how the absence of Yen1 affects the accumulation of replication and recombination across multiple cell cycles is difficult to predict. In preliminary experiments with *dna2-2 yen1Δ*, we have not observed

any dramatic increase of such intermediates above the levels seen in *dna2-2* single mutant cells. This can be rationalized considering that chromosome entanglements resulting from Dna2 helicase dysfunction might be purged from the population of cycling cells at cell division regardless of the presence – in which case they are resolved – or absence of Yen1, where they must disintegrate during chromosome segregation or when cytokinesis occurs.